# PIF transcriptional regulators are required for rhythmic stomatal movements

Arnau Rovira [1,7], Nil Veciana [1,7], Aina Basté-Miquel[1], Martí Quevedo[1], Antonella Locascio [2,6], Lynne Yenush [2], Gabriela Toledo-Ortiz[3], Pablo Leivar [4] & Elena Monte [1,5] ✉

Stomata govern the gaseous exchange between the leaf and the external atmosphere, and their function is essential for photosynthesis and the global carbon and oxygen cycles. Rhythmic stomata movements in daily dark/light cycles prevent water loss at night and allow $CO_2$ uptake during the day. How the actors involved are transcriptionally regulated and how this might contribute to rhythmicity is largely unknown. Here, we show that morning stomata opening depends on the previous night period. The transcription factors PHYTOCHROME-INTERACTING FACTORS (PIFs) accumulate at the end of the night and directly induce the guard cell-specific $K^+$ channel *KAT1*. Remarkably, PIFs and KAT1 are required for blue light-induced stomata opening. Together, our data establish a molecular framework for daily rhythmic stomatal movements under well-watered conditions, whereby PIFs are required for accumulation of KAT1 at night, which upon activation by blue light in the morning leads to the $K^+$ intake driving stomata opening.

After germination, seedlings exposed to light undergo photomorphogenesis and rapidly gain photosynthetic capacity with the separation and expansion of cotyledons, and the development of functional chloroplasts and functional stomata, the cellular structure that controls gas exchange in plants[1,2]. Photomorphogenesis is characterized by intense transcriptional reprogramming/activity[3] driven by light-activated photoreceptors, which in *Arabidopsis thaliana* include the red/far-red light-sensing phytochromes (phy), the blue/UV-A sensing cryptochromes (cry), phototropins (phot), and ZEITLUPE/ FLAVIN-BINDING, KELCH REPEAT, F-BOX1/ LOV KELCH PROTEIN 2 (ZTL/ FKF1/LKP2), and the UV RESISTANCE LOCUS8 (UVR8) receptor for UV-B[4]. Transcriptional regulators like the basic helix-loop-helix domain-containing PHYTOCHROME-INTERACTING FACTORS PIFs have a central role in this process. PIFs accumulate and act as repressors of photomorphogenesis in the dark, whereas in the light active phytochromes induce their rapid degradation via direct physical interaction[5,6]. PIFs are also involved in the regulation of the diurnal rhythm of elongation[7–10].

Stomata consist of two specialized epidermal guard cells (GC) that govern the gaseous exchange between the leaf and the external atmosphere[11]. Stomatal function is therefore essential for plant photosynthesis, respiration, and overall plant productivity, and it contributes to the global carbon cycle and replenishment of atmospheric oxygen[12,13]. Stomatal movement is determined by the osmotic concentration and the water uptake of GC. Swelling or shrinking of the GC (leading to stomata opening and closing, respectively) are driven by the activity of ion channels and ion transporters found in the plasma and vacuolar membranes of GC[14]. During stomatal opening, a plasma membrane $H^+$-ATPase pump mediates $H^+$ efflux from the cytosol and hyperpolarizes the membrane, triggering the activation of voltage-regulated inward-rectifying $K^+$ channels and leading to $K^+$ influx, with $Cl^-$ and malate as counterions. Transport of $K^+$, $Cl^-$ and malate into the

[1]Centre for Research in Agricultural Genomics (CRAG) CSIC-IRTA-UAB-UB, Campus UAB, Bellaterra, Barcelona, Spain. [2]Instituto de Biología Molecular y Celular de Plantas (IBMCP), Universitat Politècnica de València-Consejo Superior de Investigaciones Científicas, Valencia, Spain. [3]James Hutton Institute, Cell and Molecular Sciences, Errol Road Invergowrie, Dundee, UK. [4]Laboratory of Biochemistry, Institut Químic de Sarrià (IQS), Universitat Ramon Llull, Barcelona, Spain. [5]Consejo Superior de Investigaciones Científicas (CSIC), Barcelona, Spain. [6]Present address: Department of biomedical science, Faculty of Health Sciences, Universidad CEU Cardenal Herrera, Alfara del Patriarca (Valencia), Spain. [7]These authors contributed equally: Arnau Rovira, Nil Veciana. ✉e-mail: elena.monte@cragenomica.es

GC vacuole decreases GC water potential, which leads to water uptake into the vacuole, turgor increase, and opening of the stomatal pore. During stomata closure, the H⁺-ATPase is inhibited and anion channels are activated, mediating the efflux of Cl⁻ and malate, membrane depolarization, and efflux of K⁺ through outward-rectifying K⁺ channels, which leads to reduced turgor and stomatal closure[13–16].

Optimal stomatal apertures are determined by the integration of different environmental and internal cues[13,17,18]. In general, closing occurs in the dark, and under water stress in response to abscisic acid (ABA), whereas light induces stomata opening, partly through inducing degradation of ABA[19,20]. This has been proposed to contribute to diurnal stomata movements[21,22], whereby stomata are open during the day and closed during the night. Although timing of stomata dynamics is critical for optimal photosynthesis and to prevent water loss, the underlying mechanisms for diurnal stomatal movements are still largely uncharacterized[23].

ABA is one of the strongest signals that drive stomatal closure[15]. In the absence of ABA, PP2C phosphatases ABA insensitive 1 (ABI1) and 2 (ABI2) are active and inhibit positive regulators for stomatal closure such as the protein kinases open stomata 1 (OST1), or the calcium-dependent protein kinases (CPKs). In the dark and/or in drought stress, ABA accumulates and triggers changes in the expression of PYR/PYL/RCAR ABA receptors and the formation of a complex between them and the PP2C phosphatases, suppressing ABI1 and ABI2 leading to the activation of OST1, CPKs and GHR1[15,24]. The active kinases then activate the anion channels, leading to anion efflux, depolarization of the guard cell plasma membrane and activation of K⁺-efflux channels. ABA also inhibits the KAT1 inward-rectifying K⁺ channel through selective endocytosis and sequestration[25], and the plasma membrane H⁺-ATPase through ABI/OST1[26]. Many of the genes that encode relevant GC transport proteins are also ABA-regulated at the transcript level[27], like the H⁺-ATPase AHA1/OST2, the inward-rectifying K⁺ channels POTASSIUM CHANNEL IN ARABIDOPSIS THALIANA 1 (KAT1) and KAT2, the SUGAR TRANSPORTER 1 (STP1), the endosomal Na⁺/H⁺ antiporter CATION/H⁺ EXCHANGER 20 (CHX20), and the nitrate importer CHLORATE RESISTANT 1 (CHL1). These genes are all down regulated in GC following ABA treatment, suggesting that their down regulation may be an important aspect of the inhibition of stomatal opening by ABA[27]. In contrast to the well-defined effect of exogenously applied ABA on stomata closure, the role of endogenous basal ABA (present under well-watered conditions) has been less studied, and the connection between ABA signaling and the diurnal cycle under non-restrictive water conditions is still unclear.

Light is one of the most important environmental signals that induces stomatal opening[28], driven mainly by two distinct pathways induced by red and blue light respectively[16,29]. The red pathway takes place at high fluence rates, is photosynthesis dependent, and is thought to coordinate stomatal opening with photosynthesis in the mesophyll cells and the guard cells, being the primary mechanism linking stomatal behavior with demands for $CO_2$[16], although GC-specific responses to red light have also been reported[30]. This red-light response might involve light sensing by the chlorophylls and/or phytochrome B, and triggers changes in the GC osmotic potential that result in water influx and therefore stomata pore opening[16,31]. The blue-light pathway, in contrast, is GC-specific and considered to be independent of photosynthesis. This pathway involves the phototropin photoreceptors, which are activated via autophosphorylation under blue light[32]. Activated phototropins directly phosphorylate the GC-specific kinase BLUE LIGHT SIGNALLING 1 (BLUS1), which in turn transmits the signal through the type 1 protein phosphatase (PP1) and its regulatory subunit PP1 REGULATORY SUBUNIT2-LIKE PROTEIN1 (PRSL1) to activate the H⁺-ATPase, causing activation of the inward-rectifying K⁺ channels, and the influx of K⁺ leading to stomatal opening. The blue-light pathway saturates at low fluence rates and drives stomatal opening at dawn[16,29].

Importantly, whether light regulates stomata movements at the transcriptional level, and how this might contribute to stomatal dynamics under diurnal conditions through the night-day cycle, is still largely unexplored. Here, we present data showing that under diurnal cycles the transcription factors PIFs are required for blue light-induced stomatal opening in the morning. We show that the night period preceding dawn, when PIFs accumulate, is necessary for robust stomatal movements. We identify the GC-specific inward-rectifying K⁺-channel *KAT1* as a direct PIF-induced target gene, and show that mutants defective in PIFs or KAT1 are impaired in morning stomata opening under diurnal conditions. Our data establish a novel molecular framework for the rhythmic diurnal movements of stomata under non-restrictive water conditions, whereby high PIF levels at the end of the night allow for PIF-dependent accumulation of *KAT1*, which, upon activation by blue light phototropin-mediated signaling leads to the GC K⁺ intake that drives morning stomata opening.

## Results

### Stomata opening under short days oscillates and requires the night period

Stomata movements in wild-type Col-0 cotyledons were measured over 24 h during the fourth day of seedling growth from germination onwards under SD conditions (8 h light + 16 h dark; Supplementary Fig. 1) (Fig. 1a–c). At the end of the night at Zeitgeber time (ZT) ZT0, stomatal pores had a mean area of ~18 μm². Between 0–3 h after dawn (ZT0-ZT3), stomata opened and reached an aperture of ~37 μm². At the end of the light period (ZT6), stomata started to close, resulting in maximum closure 1 hour after the transition from light to dark at ZT9 (~12 μm²). During the night (ZT8-ZT24), stomata remained closed with a slight increase in stomata pore area, from to ~12 μm² at ZT9 to ~16–20 μm² at ZT24 (Fig. 1a). Similar stomatal dynamics were obtained when opening was measured using stomata width (Fig. 1b), whereas stomatal length was maintained relatively constant across the day with a slight increase between ZT0 and ZT3 (Supplementary Fig. 2). Together, these results suggest that the dark-to-light transition in SD induces stomata aperture, whereas the subsequent light period during the day promotes closing of the stomata, which remain closed during the night period. These daily oscillations in stomata movements are in accordance with and extend previous observations in older plants[33].

Next, to better understand how the diurnal light/dark cycles regulate these stomata movements, Col-0 seedlings were first grown under SD conditions for 2 days and then transferred to constant white light at the end of the third day (ZT8) for 16 h (LL conditions; Supplementary Fig. 1). Controls were kept under SD (Col-0 SD) (Fig. 1d). Stomata pore area was monitored during the fourth day every 3 h over a period of 9 h. At ZT0 and ZT9, stomata aperture was similar in Col-0 SD and Col-0 LL. However, in striking contrast to Col-0 SD, seedlings in LL were not able to open the stomata at ZT3 (Fig. 1d). Together, these results indicate that under night/day cycles, the dark period is required for stomata opening in the following morning, and suggest that one or more factors necessary for stomata opening might accumulate during the night.

### PIFs are necessary to induce stomata opening at dawn

The results in the previous section prompted us to hypothesize that PIFs, which accumulate during the night in SD, and are necessary under these SD conditions for dynamic responses such as hypocotyl elongation[7–10], might be involved in stomata movements. To test this, we measured the stomata pore in *pif* quadruple (*pifq*) mutants (deficient in PIF1, PIF3, PIF4, and PIF5) over 24 h in 4-day-old seedlings (Fig. 1a–c). No significant differences in stomata pore aperture were detected in *pifq* compared to Col-0 at the end of the night (ZT0), but strikingly *pifq* was not able to open the stomata 3 h after dawn, in clear contrast to Col-0 (~23 μm² compared to ~37 μm²) (Fig. 1a, c). Similar results were obtained when measuring stomata width (Fig. 1b), whereas stomatal length remained comparable to WT (Supplementary Fig. 2). Before the

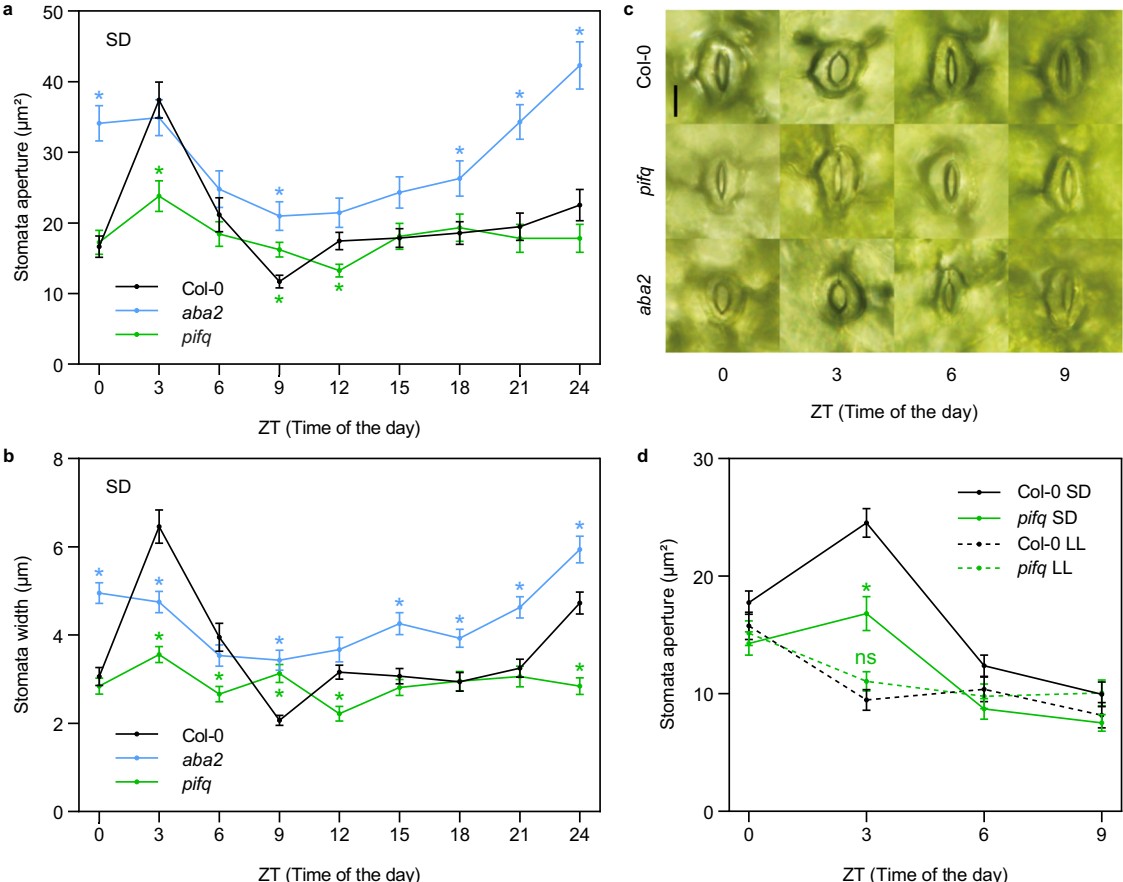

**Fig. 1 | Role of PIFs and endogenous ABA in the regulation of stomata movements in dark/light cycles. a, b** Time course analysis of stomata aperture (expressed as area in **a** and as stomata width in **b**), in Col-0, *pifq*, and *aba2* cotyledons over 24 h during the fourth day of seedlings grown under SD conditions. **c** Visible phenotypes of representative stomata in **a** and **b** at ZT 0, 3, 6, and 9h. Bar = 10 μm. **d** Time course analysis of stomata aperture (expressed as area) in Col-0 and *pifq* grown under short days (SD) or SD transferred to continuous light (LL) (see Supplementary Fig. 1 for a diagram of light treatments). Seedlings were grown under SD conditions for 2 days, then at ZT8 of the third day they were either kept under SD as a control (SD) or they were transferred to continuous white light (LL). Stomata measurements were performed during the fourth day at ZT0, 3, 6, and 9h. **a, b, d** Time points represent mean values ± SE. *n* = 40–60 biologically independent samples (precise *n* values for each genotype and time point are provided in the Source Data file). Statistical differences relative to Col-0 (**a, b**) for each time point, or **d** for each time and condition, are indicated by an asterisk (Mann–Whitney test. *P* < 0.05). Precise *P* values are provided in the Source Data file.

light-to-dark transition (ZT6), Col-0 started to close the stomata, reaching the same aperture as *pifq*. Importantly, under LL conditions, where PIF3 and likely other PIFs do not accumulate[8,9], Col-0 was not able to open the stomata in the morning, similarly to *pifq* in SD (Fig. 1d). These results led us to conclude that under SD, PIFs are required for morning stomata opening. In addition, Col-0 seedlings grown for 3 days under SD conditions and then transferred to an extended night (DD conditions; Supplementary Fig. 1), did not open their stomata in the subjective morning (Supplementary Fig. 3). Together, our data indicate that under diurnal conditions, morning stomata opening requires (1) PIF accumulation during the previous night, and (2) the transition to light in the morning. This indicates that darkness during the night period (to allow for PIF accumulation) and the transition to light at dawn, are both necessary for stomata movements under diurnal conditions. Importantly, stomatal dynamics in *pif* single mutants was similar to Col-0 (Supplementary Fig. 4a), indicating that PIFs likely control stomata dynamics in a redundant manner. Stomatal opening in PIF over-expressing seedlings was also similar to Col-0 (Supplementary Fig. 4b), suggesting tight regulation of stomata dynamics. In agreement with previous results[31] we found that stomata opening was impaired in a *phyAB* mutant (Supplementary Fig. 5), suggesting that phy-mediated control of stomata dynamics may involve regulatory mechanisms beyond the regulation of PIF accumulation, such as through the impact on photosynthesis[34].

## Endogenous basal ABA prevents early stomata opening during the night under SD

ABA promotion of stomata closure is well established under drought conditions[21] or when applied exogenously[35], whereas the contribution of endogenous basal ABA to diurnal stomata dynamics has been less studied. To address ABA's role under well-watered diurnal conditions, we measured stomata pore in the ABA-biosynthesis-deficient mutant *aba2* under our diurnal setup[36]. At ZT3, *aba2* stomata were open to an extent similar to Col-0 (~34 μm²), and were closed at the end of the light period, although not as much as Col-0 (~21 μm² compared to ~12 μm²). Remarkably, in sharp contrast to the WT, *aba2* stomata then opened progressively during the night to display fully open stomata at ZT24 (reflected in pore area, width, and length) (Fig. 1a–c, Supplementary Fig. 2). These results suggest that the main role of basal ABA under SD conditions is to maintain the stomata closed during the night hours. This finding is consistent with previous data showing that endogenous ABA accumulates to a maximum during the dark period[19,37], and with results showing that ABA-insensitive plants display more open stomata in the dark[38]. In addition, our results show that endogenous basal ABA is necessary for full stomatal closure during dusk, although other factor(s) are probably involved given that *aba2* seedlings at ZT9 display more open stomata than Col-0 (~21 μm² compared to ~12 μm²; Fig. 1a–c). Interestingly, in LL conditions, *aba2* stomata remained completely open during ZT0–ZT9, in contrast to SD

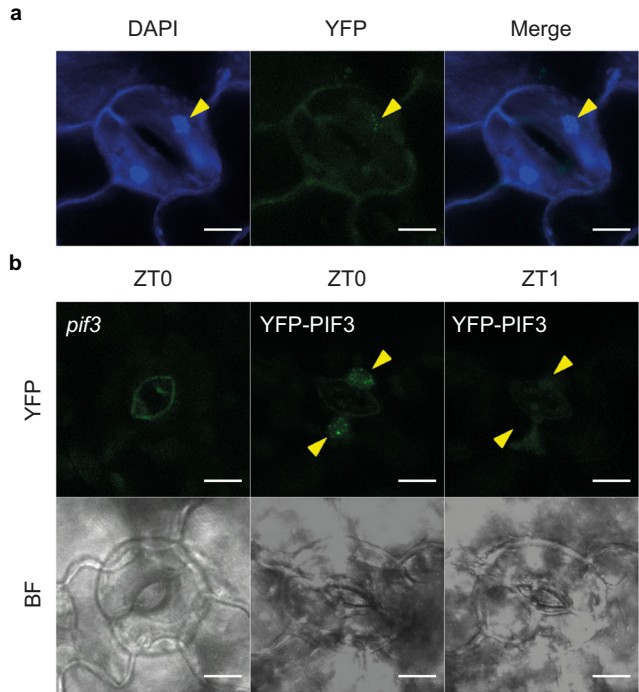

**Fig. 2 | PIF3 localizes in the nucleus of guard cells at the end of the night and is rapidly degraded by light. a** YFP-PIF3 localization in guard cells of 3-day-old SD-grown pPIF3::YFP-PIF3 Arabidopsis seedlings (ZT23). DAPI was used for nucleic acid staining. **b** YFP-PIF3 localization in guard cells of 3-day-old SD-grown (ZT23) and 4-day-old SD-grown (ZT1) pPIF3::YFP:PIF3 Arabidopsis seedlings after 1 h of white light. Images correspond to the same stomata. pif3 was used as a control (signal corresponds to autofluorescence). Bar = 10 μm. Bright field (BF) images are shown. The experiments were repeated twice with similar results.

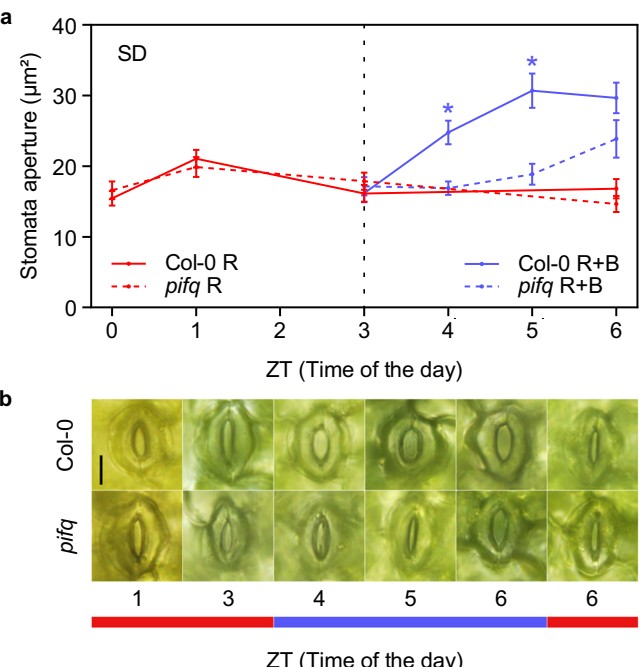

**Fig. 3 | PIFs promote blue-light induce stomata opening in the morning of dark/light cycles. a** Time course analysis of stomata aperture in 3-day-old SD-grown Col-0 and pifq seedlings exposed to 3h of red light (40 umols/m2.s). In this background, blue light (10 umols/m2.s) was added for an additional 3h. Controls were kept in red light for the whole duration of the experiment. Time points represent mean values ± SE. $n = 56–90$ biologically independent samples (precise n values for each genotype and time point are provided in the Source Data file). Statistical differences relative to Col-0 for each time point and condition are indicated by an asterisk (Mann–Whitney test. $P < 0.05$). Precise P values are provided in the Source Data file. **b** Visible stomata phenotypes of representative seedlings grown in **a**. The red and blue bar indicate the light under which the samples were taken. Bar = 10 μm.

and also to DD, suggesting that the unknown factor(s) acting together with ABA at dusk might not accumulate or be active in LL (Supplementary Fig. 6).

### PIF3 localizes in the nucleus of guard cells at the end of the dark phase under SD and is degraded by light at dawn

Our analysis suggests that the previously described accumulation of PIFs at the end of the night under diurnal conditions[7–9] is required for the subsequent stomata aperture at dawn. To address whether PIF-mediated stomata opening activity could be done locally, we next study whether PIF3 accumulates in guard cells. Visualization of 3-day-old SD-grown pPIF3::YFP:PIF3 (YFP-PIF3) at the end of the night (ZT0) by confocal microscopy showed PIF3 in the nucleus of guard cells, localized in the characteristic speckles (Fig. 2a)[39]. Notably, YFP fluorescence was almost undetectable after 1 hour of white light treatment (ZT1) (Fig. 2b). Together, these results indicate that PIF3 accumulates in guard cells at the end of the dark period and is rapidly degraded by light. This pattern of PIF3 protein accumulation in the dark and its degradation upon light exposure is in accordance with the described presence of phytochrome in guard cells[40,41], and it is comparable to that described in hypocotyl cell nuclei of seedlings from the same pPIF3::YFP:PIF3 line[42], or in PIF3 overexpressing seedlings[39].

### PIFs are necessary for rapid blue-light-induced stomatal opening

Light-induced stomatal opening is driven mainly by two distinct pathways induced by red and blue light, respectively[16,29]. The red or mesophyll/photosynthetic pathway takes place at high fluence rates and is thought to coordinate stomatal opening with photosynthesis.

The blue-light pathway, on the other hand, is phototropin-mediated, considered guard cell-specific and independent of photosynthesis, saturates at low fluence rates and drives stomatal opening at dawn. To further define the role of PIFs in stomatal opening and understand PIF contribution in the red and/or blue pathways, we next performed an experiment following the set up reported by Papanatsiou et al.[43]. Col-0 and pifq SD-grown seedlings were first exposed to 3 h of red light (40 μmols/m²s) at dawn to provide a photosynthetic energy input that reduces $CO_2$ concentration[43]. We observed partial transient stomata opening in Col-0 after 1 h of red light treatment (ZT0 to ZT1; from 18 to 25μm²), and no significant differences in stomatal opening between Col-0 and pifq (Fig. 3). At ZT3, after 3h of red light, pore areas in Col-0 and pifq stomata had closed again similar to ZT0. In this background, adding blue light (10 μmols/m²s) rapidly (in 2 h) elevated stomata opening in Col-0 from 18 to 33 μm². Remarkable, this response was completely absent in pifq (Fig. 3). After 3h of blue light treatment, Col-0 stomata continued open (32 μm²) indicating that the stomata response was already at a maximum after 2 h, whereas pifq started opening between ZT5 and ZT6. Controls in red light remained closed (Fig. 3). Together, these results indicate that PIFs are necessary for fast blue light stomatal opening, and strongly suggest that the impaired stomatal opening in pifq in the morning is a consequence of a deficiency in the blue light pathway. Given the described PIF accumulation during the night hours and the rapid phytochrome-mediated degradation upon light exposure[8], we hypothesize that PIFs might be necessary to regulate the expression during the previous night of a key component in the guard-cell specific phototropin pathway leading to stomatal opening in the morning.

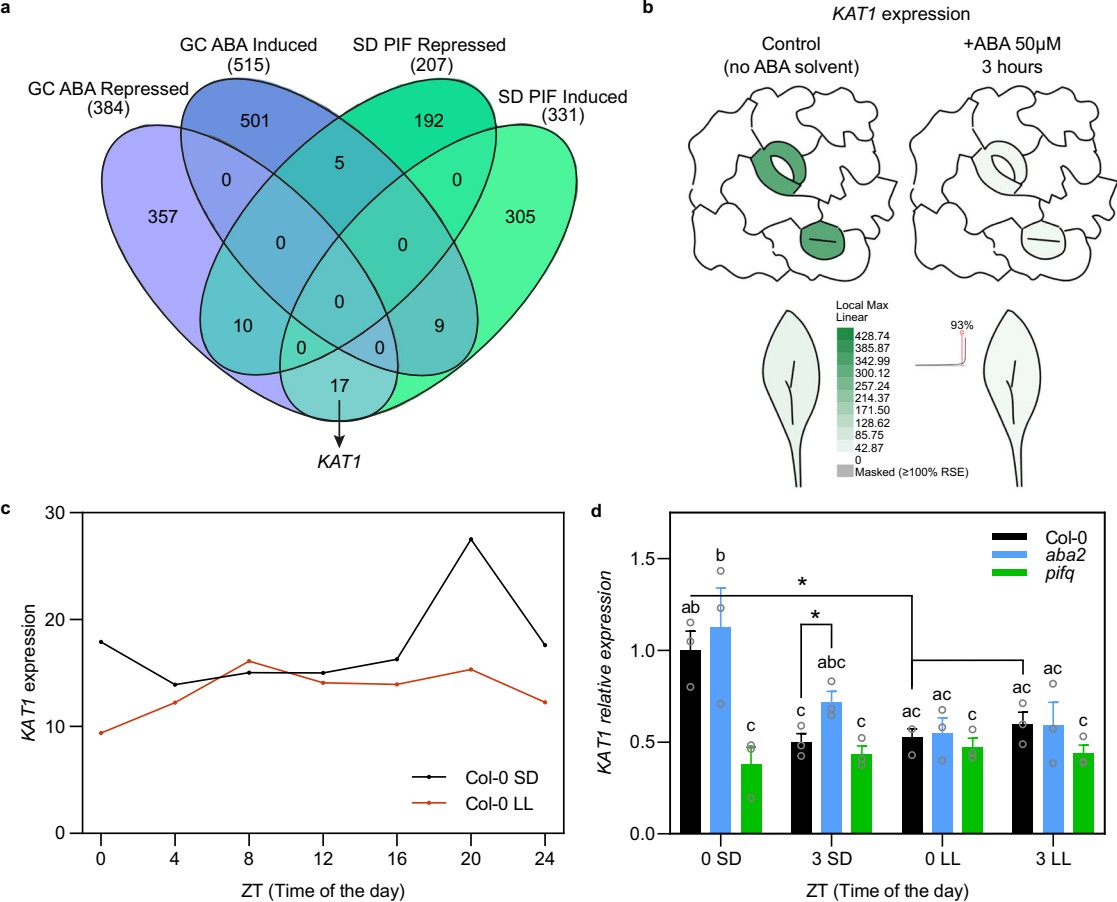

**Fig. 4 | Identification of *KAT1* as a guard-cell specific gene induced by PIFs and repressed by exogenous ABA. a** Venn diagram depicting total numbers (in parentheses) and overlap of PIF-regulated genes in short days (SD)[44], and guard cell (GC) specific ABA-regulated genes[27]. The comparison identified *KAT1* in the gene set comparison indicated by the arrow. **b** *KAT1* expression in Col-0 guard cells and leaves in SD-grown plants. Plants treated 3 h with 50 μM of ABA or solvent control are shown. Data obtained from ePlant (https://bar.utoronto.ca/eplant/)[14]. **c** Time course analysis of *KAT1* expression in Col-0 under SD and under continuous light condition after entrainment in 12:12 (LL). Data obtained from DIURNAL5 (http://diurnal.moclker.org)[83]. **d** *KAT1* expression in 3-day-old SD- or LL-grown Col-0, *pifq*, and *aba2* seedlings at ZT0 and ZT3. Data are the means ± SE of biological triplicates (*n* = 3). Letters denote the statistically significant differences using 2-way Anova followed by posthoc Tukey's test (*P* < 0.05), and asterisks indicate statistically significant differences between specific samples (*t* test; **P* < 0.05). The precise *P* values are provided in the Source Data file.

## Identification of the inward-rectifying potassium channel *KAT1* as a PIF-induced and ABA-repressed guard cell-specific gene under SD conditions

To explore the possibility that PIFs regulate expression of a necessary component for stomatal opening in the phototropin-mediated pathway, and given the regulation of stomata movements by PIFs and ABA, we next aimed to identify ABA-responsive genes in guard cells that might be PIF targets. We reasoned that these genes could encode proteins involved in stomatal dynamics downstream of the PIFs. Because ABA induces stomatal closure whereas PIFs are necessary for aperture, we hypothesized that relevant PIF-regulated genes in stomatal dynamics could correspond to GC genes inversely regulated by PIFs and ABA. To this end, we compared previously defined gene sets of PIF-regulated genes in 3-day-old SD-grown seedlings at the end of the night (538 genes, 331 induced and 207 repressed by PIFs)[44] with ABA-responsive guard-cell specific genes (906 genes, 515 induced and 384 repressed by ABA) (identified in isolated guard cells but not in whole leaf, and responsive to a treatment of 50 μM of exogenous ABA for 3 h)[27]. Forty-one common genes were identified, and of these, 22 were inversely regulated by PIFs and ABA (Fig. 4a and Supplementary Data 1). Interestingly, one of these genes (*AT5G46240*) is *KAT1*, encoding the voltage-dependent potassium channel KAT1 predominantly expressed in GC (Fig. 4b). KAT1 mediates the potassium influx that leads to stomata swelling and opening[14,45,46], through a unique gating mechanism that is activated upon the membrane hyperpolarization triggered by phototropin-mediated blue light signaling[29,47]. The *KAT1* gene is induced by PIFs and repressed by ABA (Fig. 4a, b): *KAT1* regulation showed a ~14-fold decrease in expression in GC upon treatment with ABA[27], whereas *KAT1* expression was 2.3-fold higher at ZT24 in WT compared to *pifq*[44] (Supplementary Data 1). In addition, publicly available diurnal data (http://diurnal.mocklerlab.org) showed a peak of *KAT1* expression in SD at the end of the night. This peak is absent in LL conditions (Fig. 4c), an expression pattern that is characteristic of PIF-regulated genes under diurnal conditions[48]. To test the possibility that in our conditions PIFs induce *KAT1* expression before dawn, we checked *KAT1* expression in WT and *pifq* at ZT0, and found that indeed PIFs are necessary for the elevated expression of *KAT1* at the end of the night. After 3 h of light at ZT3, expression levels in the WT decreased (Fig. 4d and Supplementary Fig. 7) to levels similar to *pifq* at ZT0 and ZT3 (Fig. 4d). Importantly, in LL *KAT1* levels at ZT0 were similar to levels in SD at ZT3 (Fig. 4d), whereas in DD levels remained high at ZT3 (Supplementary Fig. 8). Together, these results indicate that PIFs induce *KAT1* expression at the end of the night, whereas the transition to light triggers a rapid decrease in *KAT1* transcript levels that correlates with light-induced PIF degradation. Interestingly, *KAT1* expression does not significantly

increase in PIF-OX seedlings (Supplementary Fig. 4c) or in *phyAB* (Supplementary Fig. 5), suggesting tight regulation of *KAT1* levels. Regarding the regulation of *KAT1* expression by ABA, although the repression by application of exogenous ABA is well defined, we wanted to test to what extent endogenous ABA in our conditions might contribute to the transcriptional regulation of *KAT1*. To this end, we analyzed *KAT1* expression in *aba2* mutant seedlings grown in SD and LL. Under SD conditions, no significant differences in *KAT1* expression were observed between Col-0 and *aba2* at the end of the night period (ZT0), whereas at ZT3 the *KAT1* levels in *aba2* were slightly more elevated compared to Col-0, differences that were not detected in LL (Fig. 4d) or DD (Supplementary Fig. 8). These data suggest that endogenous basal ABA does not significantly repress PIF-mediated induction of *KAT1* expression at night, and might contribute to the repression of *KAT1* expression in SD upon exposure to light. This is in contrast to the effect of exogenously applied ABA, which significantly repressed *KAT1* expression as expected, and concomitantly prevented light-induced stomata opening in the morning (Supplementary Fig. 9). Taken together, we conclude that in SD PIFs are necessary to induce the expression of *KAT1*, a GC specific gene that encodes the inward-rectifying K⁺ channel driving the K⁺ uptake that leads to stomata opening upon activation by blue light. The induction of *KAT1* is dark-dependent and peaks at the end of the night, whereas light exposure leads to a repression in *KAT1* expression, likely as a result of PIF degradation with a minor contribution of endogenous ABA.

## *KAT1* expression is directly regulated by PIFs at the end of the dark period under SD

We next addressed whether *KAT1* might be a direct PIF target. We found several G-box and PBE binding motifs approximately 1 kb upstream of the transcription start site (TSS) in the *KAT1* promoter that could act as potential PIF binding sites (Fig. 5a). Indeed, available chromatin immunoprecipitation (ChIP)-seq data[49] showed binding of PIF1, PIF3 and PIF4 (statistically significant for PIF1 and PIF4) to this region in 2-day-old dark-grown seedlings (Fig. 5a). To test whether PIFs could bind to the *KAT1* promoter in our diurnal conditions, we performed ChIP followed by qPCR (ChIP-qPCR) analysis at the end of the night period (ZT0) using the PIF3-tagged line driven by the endogenous PIF3 promoter (YFP-PIF3) (the same line used previously in Fig. 2) and a PIF4-tagged line (PIF4-HA) driven by the 35S promoter. After ChIP, primer pairs P1 and P2, encompassing the G-boxes and PBE in this region, were used for qPCR. Statistically significant YFP-PIF3 and PIF4-HA binding was observed for P2 (Fig. 5b) and for P1 and P2 (Fig. 5c) respectively, whereas a primer combination (P3) that binds to the last exon was used as a negative control (Fig. 5b, c). A yeast one hybrid assay (Y1H) confirmed specific binding of PIF3 to the P2 region of the *KAT1* promoter through the G-box motif (Supplementary Fig. 10). We conclude that PIF3 and PIF4 directly bind to the *KAT1* promoter under diurnal conditions.

## *KAT1* is essential to induce morning stomata opening

Previous reports have described contrasting results as to whether KAT1 deficiency might be enough to impair stomata movements[50–52]. To evaluate the relevance of the PIF-KAT1 module to mediate stomata dynamics in our conditions, we obtained two independent mutant lines lacking *KAT1* (*kat1-1* and *kat1-2*)[53]. Stomata aperture analyses showed that at the end of the night (ZT0), *kat1-1* and *kat1-2* displayed slightly more closed stomata compared to Col-0 (13 µm² compared to 20 µm²). Strikingly, 3 h after dawn (ZT3), *kat1-1* and *kat1-2* stomata remained closed, in contrast to Col-0, which displayed fully opened stomata (33 µm²). At ZT6, Col-0 seedlings had closed stomata, similar to *kat1-1* and *kat1-2*. No differences were observed at ZT9 (Fig. 6a). These results indicate that, under day/night cycles, KAT1 is essential for stomata opening in the morning. Next, we tested a *KAT1OX* line.

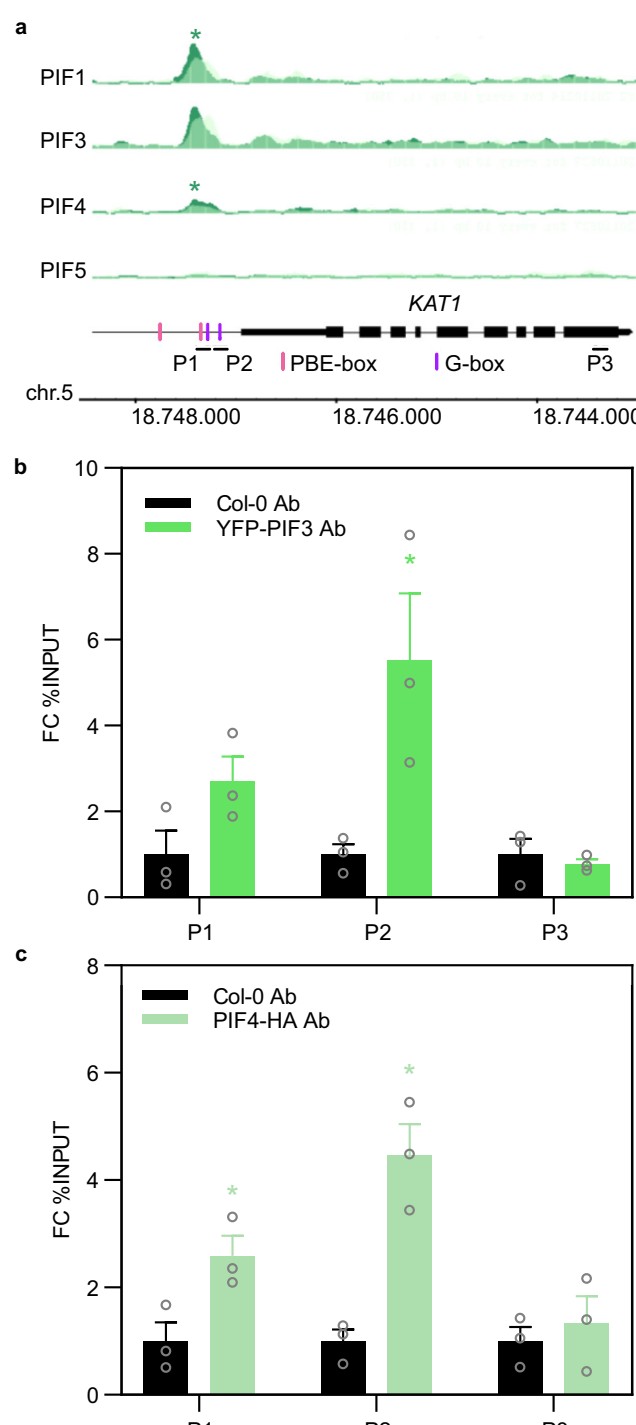

**Fig. 5 | *KAT1* is a PIF direct target. a** Visualization of ChiP-Seq data obtained for PIF1, PIF3, PIF4, and PIF5 in the genomic region encompassing the *KAT1* locus (dark green tracks). Overlaid light green tracks indicate the corresponding WT binding control. Identified significant binding sites are indicated by an asterisk on top of the pile-up tracks. Data obtained from[49]. G-box (CACGTG) and PBE-box (CACATG) motifs in the *KAT1* promoter region are indicated. **b**, **c** ChIP-qPCR binding of YFP-PIF3 (**b**) and PIF4-HA (**c**) to the P1 and P2 regions in *KAT1* promoter at ZT24 in 3-days-old SD-grown seedlings. PIF binding is represented as fold change (FC) % of input and relative to Col-0 set at unity. Col-0 and the intergenic primer P3 were used as negative controls. Bars represent mean values ± SE. *N* = 3 biological replicates. Statistical differences between mean log₂ FC values relative to Col-0 for each pair of primers are indicated by an asterisk (Student *t* test. *P* < 0.05). Precise *P* values are provided in the Source Data file.

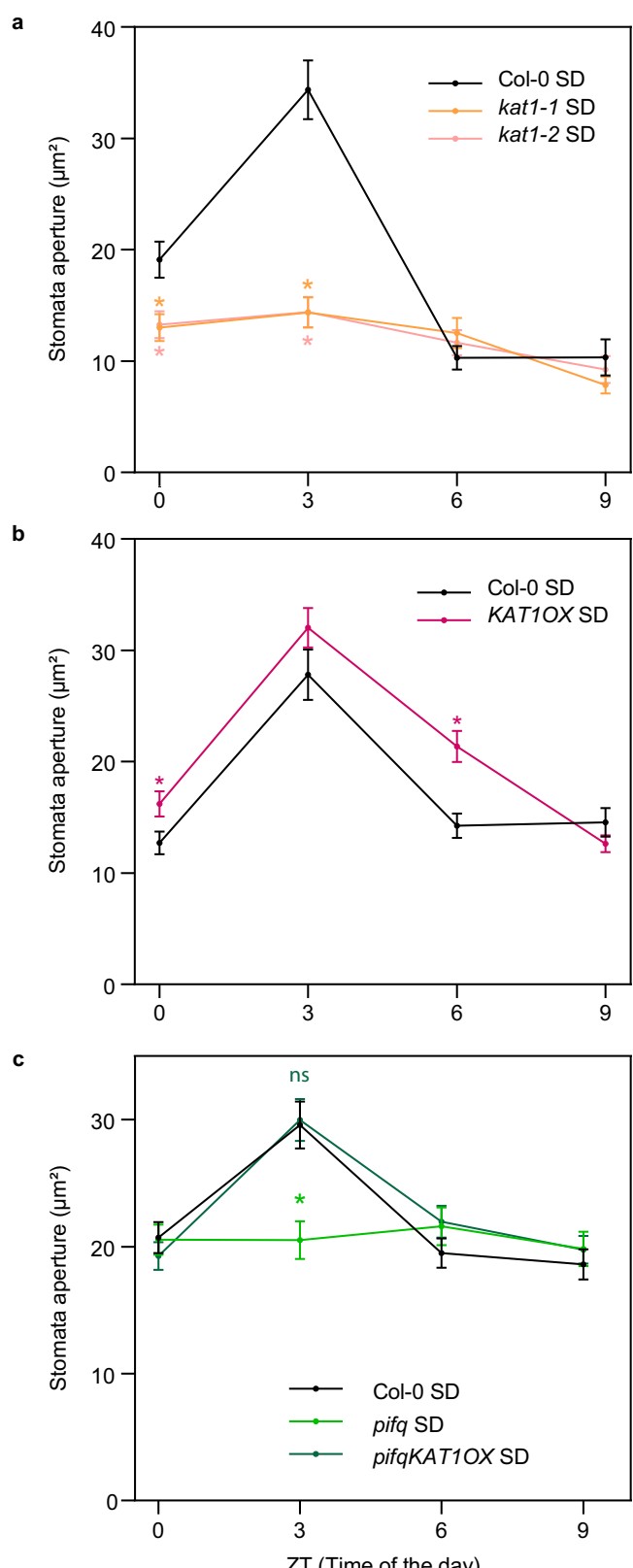

**Fig. 6 | KAT1 regulates stomata opening in the morning in dark/light cycles.**
**a, b** Time course analysis of stomata aperture in (**a**) Col-0 and two *kat1* alleles (*kat1-1* and *kat1-2*), (**b**) Col-0 and *KAT1OX*, and (**c**) Col-0, *pifq* and *pifqKAT1OX*, during the fourth day of seedlings grown under SD conditions. Time points represent mean values ± SE. *n* = 40–107 biologically independent samples (precise *n* values for each genotype and time point are provided in the Source Data file). Statistical differences relative to Col-0 for each time point are indicated by an asterisk (Mann–Whitney test. *P* < 0.05). Precise *P* values are provided in the Source Data file.

Compared to Col-0, *KAT1OX* had slightly more open stomata in the dark at ZT0, reached similar light-induced stomata opening at ZT3, and displayed slower stomata closing at the end of the day (ZT6). At ZT9, *KAT1OX* stomata were as closed as the Col-0 (Fig. 6b). Interestingly, overexpression of *KAT1* displayed slightly more opened stomata in the non-inductive LL conditions (Supplementary Fig. 11). Together, these data indicate that under diurnal conditions: (1) alteration in KAT1 levels had a small but significant effect on stomata movements at the end of night; (2) KAT1 is required to induce stomata opening at dawn; and (3) ectopic expression of KAT1 resulted in similar light-induced stomata pore area that remained opened for longer time during the day period, an opening effect that was also observed in dark and LL. Finally, to provide conclusive support for the PIF-KAT1 module in the regulation of stomata dynamics, we generated *pifqKAT1OX* lines and tested stomatal opening at the end of the night and during the first hours of light exposure. Remarkably, KAT1 overexpression was able to fully restore stomata aperture in *pifq*, and *pifqKAT1OX* lines displayed stomata opening similar to WT in both white light (Fig. 6c and Supplementary Fig. 12a) and in response to blue light (Supplementary Fig. 12b).

## Discussion

We have shown here that the phytochrome-interacting transcription factors PIFs are required for establishing the rhythmicity in stomatal dynamics under diurnal conditions, through direct induction of the expression of *KAT1*, a guard-cell specific gene encoding an inward-rectifying potassium channel that is necessary for stomatal opening. Our work unveils a novel regulatory link between light/dark cycles and stomata movements, whereby PIF accumulation during the night results in *KAT1* transcript accumulation at dawn, in preparation for stomata opening in the morning (Fig. 7 and Supplementary Fig. 13). In the beginning of a new day, when the blue/red light ratio is high, light triggers phototropin-mediated blue-light induced hyperpolarization of the membrane, which drives ion influx through KAT1, leading to water intake, turgor increase and stomata opening[13]. Light also induces phytochrome-mediated PIF degradation, which ensures that *KAT1* induction is not maintained during the day, which could result in slower stomata closure as seen with KAT1OX lines (Fig. 6). Moreover, using the ABA-deficient mutant *aba2*, our results show that under diurnal cycles in non-restrictive water availability, endogenous ABA levels are necessary for full dark-induction of stomata closure, and are required to prevent stomata opening during the night (Fig. 7 and Supplementary Fig. 13).

These results, together with previous data, support a model whereby a novel interplay between PIF function and endogenous ABA signaling in the guard cell establishes the dynamic regulation of stomata aperture under diurnal cycles: at dusk and in the dark, endogenous ABA is produced[21], which induces stomata closure and is necessary to maintain stomata closed during the night. Before dawn, accumulated PIFs induce *KAT1* expression (Fig. 7 and Supplementary Fig. 13). ABA likely prevents KAT1 accumulation at this time through post-transcriptional mechanisms such as KAT1 endocytosis and sequestration[21]. In the morning, a reduction in ABA is necessary for light-induced stomata opening[54], by means of light-induced depletion of endogenous bioactive ABA through inactivation and catabolism coupled with a reduction in ABA biosynthesis[22,55]. In the absence of ABA, KAT1 accumulates and light-induced phototropin-mediated guard cell membrane hyperpolarization leads to KAT1 activation and GC opening. In the absence of PIFs in *pifq*, *KAT1* is not expressed before dawn and lack of KAT1 prevents morning stomata opening, which can be restored in *pifqKAT1OX* (Fig. 6c). Thus, morning stomata dynamics in diurnal conditions requires two distinct processes to occur: (1) accumulation of PIFs during the night to induce *KAT1* expression, and (2) light-induced inactivation of ABA to allow KAT1 protein accumulation.

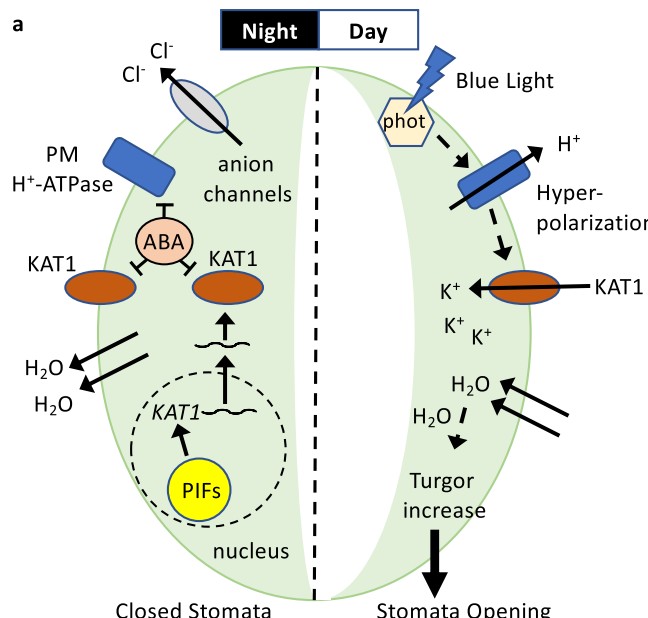

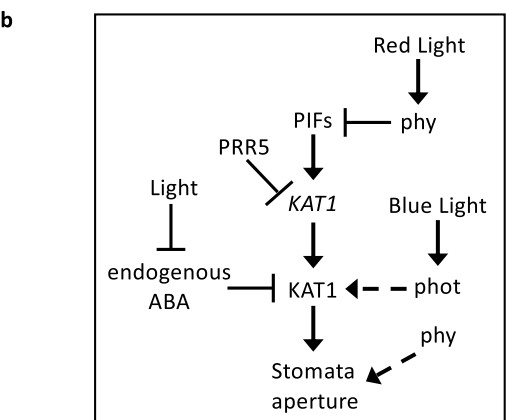

**Fig. 7 | Model of PIF-mediated regulation of stomatal movements. a** Guard cell (GC) cartoon and (**b**) schematic model depicting PIF-mediated regulation of stomatal movements in the dark during the night and during the day. At night, PIFs accumulate and induce *KAT1* transcription. Endogenous ABA represses activity of KAT1, as well as that of the plasma membrane H+ pump (PM H+-ATPase). Anion and K+ efflux reduces the GC turgor causing stomata to close. At dawn, blue light activates phototropins, which initiate a signaling cascade to activate the plasma PM H+-ATPase that transports H+ across the membrane, causing a hyperpolarization that activates the KAT1 channel and induces an influx of K+ and accumulation of K+ and counteranions (Cl− and malate) into the GC and its vacuole. Accumulation of these ions leads to water uptake into the vacuole and turgor increase, triggering stomatal opening. In the morning, red light activated phytochromes degrade PIFs and prevent *KAT1* overexpression. Phytochromes can also impact stomata aperture through alternative pathways. Through a yet unknown mechanism, PRR5 can repress KAT1 expression.

PIF-promoted rhythmicity of stomata movements is reminiscent of the well-established role of PIFs in promoting rhythmic hypocotyl elongation under diurnal conditions of short days, where growth rate reaches a maximum at dawn, a response that is PIF dependent[7–10]. Both PIF-regulated responses (stomata opening and hypocotyl elongation) share some striking similarities: (1) Both occur at dawn with the requirement of a previous night period (Fig. 1)[8]; (2) Both are promoted by PIFs (Fig. 1)[9]; (3) In both, timing at dawn likely correlates with the highest PIF levels throughout the day/night cycle, as determined at whole-seedling level[7,8] and here in guard cells during the night-to-day

transition (Fig. 2), in accordance to the described pattern of PIF accumulation in the dark and light-induced degradation[39,42]; And (4), in both, PIF-mediated transcriptional regulation of key components underlies the response, shown here for *KAT1* and stomata movement (Figs. 4−6), and previously for gene networks involved in cell elongation and hypocotyl growth (like auxin or cell wall remodeling)[44,48,56,57]. Interestingly, we and others previously demonstrated that under diurnal conditions, PIF activity during the night is gated to pre-dawn by the circadian clock components and transcriptional repressors Timing of CAB expression 1 (TOC1 or pseudo response regulator (PRR) 1), PRR5, 7 and 9, by direct interaction and co-binding with PIFs on the G- and PBE-box DNA motifs found in target genes involved in cell elongation[48,58–60]. Whether the clock contributes to gate PIF activation of *KAT1* induction is currently unknown. Interestingly, ChIP-seq experiments with PRR proteins identified binding of PRR5 to *KAT1*[61,62], and we found that indeed PRR5 negatively regulates *KAT1* expression under our diurnal conditions (Supplementary Fig. 14). This transcriptional repressing activity of PRR5 could potentially be part of a tight regulatory mechanism of *KAT1* expression, and might explain why elevated levels of PIF in *PIFOX* or *phyAB* (Supplementary Figs. 4 and 5) did not lead to significant *KAT1* overexpression. Further investigation will be required to understand the role of PRR5 in the optimization of stomata dynamics.

The diurnal setup used here favored blue-light mediated stomata opening in the morning, and this has allowed us to uncover a novel role for PIFs as promoters of stomata opening under these conditions. Interestingly, recent studies have reported a role for PIFs as negative regulators of stomata opening under red light[31,63]. Although we did not detect this effect in our experiments in red light (Fig. 3), this might be explained by developmental effects (since older plants were used), different light conditions (such as continuous light) or a consequence of different water content (under water stress or applying exogenous ABA). Indeed, a role for the PIF-like OsPIL15 in coordinating red light and ABA signaling to regulate stomatal aperture in rice plants was reported under drought[63]. Further studies will be necessary to address the relative contribution of PIFs to red and blue light stomata opening in day/light conditions under different water content.

The requirement of KAT1 for stomatal movements was a matter of controversy in the past years. An early work published in 2001[50] using a transposon insertion mutant reported that KAT1 was not essential for stomata opening, mostly based on lack of mutant phenotype, although patch-clamp experiments in *kat1* mutant protoplasts did show strong impairment for K+ conductance compared to wild type[50]. More recently, Takahashi et al.[51] reported deficient stomata opening phenotypes for *kat1* under certain conditions. Finally, Locascio et al.[52] showed clear impairment of light-induced stomata opening in whole leaves of *kat1* mutant plants. In view of our findings here regarding the importance of the light regime and time of day to induce maximal stomata opening in the WT (essential to detect possible differences with the mutant, as exemplified by the lack of stomata phenotype in *pifq* in continuous light or during most part of the diurnal cycle (Fig. 1)), a possible explanation for these differences is that the growth conditions and/or sample preparation (for example epidermal peel manipulation in the light) could have led the authors to overlook the stomata opening phenotype of KAT1-deficient mutants.

Our findings, together with the described regulation by ABA[27], suggest that KAT1 might be a fundamental hub for the regulation of stomata dynamics. In accordance, previous reports showed that *KAT1* expression is enhanced by the ABA-regulated AKS1 transcription factor[51], by auxins[64] and by brassinosteroids[65]. Although the implication of these regulatory levels in stomata opening in day/night cycles is currently unknown, the emerging picture seems to suggest that *KAT1* can integrate information of diverse environmental and endogenous factors including photoperiod, time of day, and hormone levels (like ABA, auxin and BRs), which together would provide a mechanism for

fast stomata response to a diversity of light conditions and water contents. Interestingly, although *KAT1* expression has been well described to be repressed by ABA[27], our observation that *KAT1* levels are not significantly affected in *aba2* seedlings under non-restrictive water availability (Fig. 4d) indicates that endogenous basal ABA is likely not sufficient to interfere with PIF-mediated induction of *KAT1*. Instead, repression of *KAT1* expression by ABA may take place only when ABA levels increase, which is mimicked by the exogenous application of high ABA levels in most studies, including the transcriptomic work used to identify *KAT1*[27], and also demonstrated here under diurnal conditions (Supplementary Fig. 9). Under stress conditions, such as those of low humidity or drought, ABA accumulation could override the PIF-mediated signal by repressing *KAT1* expression to prevent morning stomatal opening and preserve water. How ABA might repress PIF-mediated induction of *KAT1* is currently unknown. Intriguingly, a recent report has shown that the ABA receptors PYR-ABACTIN RESISTANCE1-LIKE8 (PYL8) and PYL9 can physically interact with PIFs and interfere with PIF activity to regulate *ABI5* expression[66]. Future work will be necessary to evaluate whether this mechanism is active in the regulation of other genes targeted by PIFs and ABA such as *KAT1*.

In conclusion, our work proposes a conceptual framework whereby the dynamic accumulation of PIFs and endogenous basal ABA throughout the day/night cycles provides the plant with an exquisite mechanism to adjust stomata movements to the precise time of the day. Correct timing and speed of stomatal movements through the day/night cycle is critical for optimized carbon uptake, photosynthesis, water use efficiency, and control of plant physiology[23,43,67]. Therefore, understanding how the day/night cycle regulates stomatal movements can provide targets to optimize plant yield with improved water use[43]. Indeed, and to illustrate the potential relevance of our findings, a recent work has revealed the importance of K+ inward-rectifying channels in determining plant biomass production, and plant adaptation to fluctuating and stressing natural environments[68].

## Methods

### Seedling growth and measurements

*Arabidopsis thaliana* seeds used in this manuscript include the previously described *aba2/gin1-3* (*aba2*)[36], *pifq*[69], *pPIF3::YFP:PIF3* (YFP-PIF3)[42], *p35S::PIF4-HA* (PIF4-HA)[70], *pif1-1*[71], *pif3-3*[72], *pif4-2*[73], *pif5-3*[74], *PIF3-OX*[75], *PIF5-OX*[74], *phyAB*[76], *prr5-1*[77], *kat1-1* and *kat1-2*[53], and *35S:KAT1-YFP* (*KAT1OX*)[78] in Col-0 ecotype. The expression of *KAT1* in the *KAT1OX* line is approximately 12-fold higher compared to Col-0 at ZT0 in SD (Supplementary Fig. 15). *pifqKAT1OX* was generated by crossing *pifq* and *KAT1OX*. Seeds were sterilized and plated on Murashige and Skoog medium (MS) without sucrose as previously described[79]. Seedlings were stratified for 4 days at 4 °C in darkness, and then placed in short day (SD) conditions [8 h white light (85 μmol m$^{-2}$ s$^{-1}$) + 16 h dark] for 3 days at 21 °C. For photobiology experiments under different photoperiods, seedlings were either kept in SD conditions for the entire duration of the experiment or grown in SD for 3 days and then transferred to continuous white light conditions (LL, 60 μmol m$^{-2}$ s$^{-1}$) or to continuous dark conditions (DD) (see Supplementary Fig. 1). For red and blue light experiments, 3 day-old SD-grown seedlings were first transferred to red light (40 μmol m$^{-2}$ s$^{-1}$) for 3 h, and were then supplemented with 10 μmol m$^{-2}$ s$^{-1}$ of blue light. Controls were kept in red. White light in SD and LL regimes (with a ratio of Red (600–700 nm)/Blue (400–500 nm) of 1.9, Supplementary Fig. 16) was provided by PHILIPS Master TL5 HO/39W/840 lamps. Red and blue light were provided by PHILIPS GreenPower research module in deep red (maximum at 660 nm), and blue (maximum at 460 nm), respectively (Supplementary Fig. 16). Light intensity was measured with a built-in LI-COR LI-190R Quantum Sensor in the red and blue+red growth chambers (Aralab) and with a hand-held SpectraPen mini (Photon Systems Instruments) for the white light growth chamber (Aralab). Light

spectra were measured with a PG200N Spectral PAR Meter (UPRtech). The temperature in all chambers was 21–22 °C. For ABA treatments, 50 μM ABA (SIGMA) or the mock equivalent (0.05% ethanol) diluted in half-strength MS liquid media were applied in dark conditions to 3-day-old SD-grown seedlings 2 h before the start of the light cycle. For stomata pore aperture measurements, cotyledons (corresponding to 6 independent seedlings) were wet mounted with water on a microscope slide with a cover glass, and pictures of the abaxial epidermis were taken with and optical microscope (Zeiss AxioPhot DP70) with the oil immersion ×63 objective lens. Dark samples followed the same procedure as light samples except that they were wet mounted under green dim light before imaging. Stomata ($n$ = 40–110) pore area or pore width and length was measured using NIH image software (Image J, National Institutes of Health). The precise number of biologically independent samples ($n$) is provided in the Source Data file of each figure.

### Fluorescence microscopy

Stomata of 3-day-old SD-grown *pPIF3::YFP:PIF3* seedlings at ZT0, and same stomata maintained in white light for 1h after dawn (ZT1), were visualized using a confocal laser scanning microscope Olympus FV1000 (Emission window: 500nm – 660nm). *pif3* was used as a negative control. Nuclei were stained using DAPI.

### Statistical analysis

Differences in stomata area, width or length between two genotypes, or between two different light conditions for the same genotype, were analyzed by a pairwise Mann-Whitney test to assess mean differences in non-parametric data. Significantly different pairs ($P < 0.05$) were represented by asterisks. In the ChIP experiment, statistical differences between mean fold change values relative to Col-0 were $\log_2$ transformed and analyzed by Student $t$-test ($P < 0.05$) (two-sided). To identify gene expression differences taking into account global variation across all genotypes and light conditions, and given the parametric nature of the gene expression measurements, data were analyzed using two-way ANOVA. A post-hoc Tukey test was performed to identify significant differences between pairs of genotypes or light conditions, and significantly different pairs were represented by letters. In specific sample pairs, a $t$-test (two-sided) was performed and asterisks indicate statistically significant differences. The precise $p$-values for each experiment are provided in the Source Data file of each figure.

### Gene expression analysis

RNA was extracted using Mawxell RSC plant RNA Kit (Promega). 1μg of total RNA extracted were treated with DNase I (Ambion) according to the manufacturer's instructions. First-strand cDNA synthesis was performed using the NZYtech First-strand cDNA Synthesis Kit (NZYtech). 2 μl of 1:25 diluted cDNA with water was used for real-time PCR (LightCycler 480 Roche) using SYBR Premix Ex Taq (Takara) and primers at 300nM concentration. Gene expression was measured in three independent biological replicates, and at least two technical replicates were done for each of the biological replicates. *PP2A* (AT1G13320) was used for normalization[80]. Primers are listed in Supplementary Table 1.

### Chromatin Immunoprecipitation and ChIP Assays

Chromatin immunoprecipitation (ChIP) and ChIP-qPCR assays were performed as previously described[58]. ChIP was performed in the dark under green safelight. Briefly, seedlings (3g) were vacuum-infiltrated with 1 % formaldehyde and cross-linking was quenched by vacuum infiltration with 0.125 M glycine for 5 min. Tissue was ground, and nuclei purified by sequential extraction on Extraction Buffer 1 (0.4 M Sucrose, 10 mM Tris-HCL pH8, 10 mM MgCl$_2$, 5 mM ß-mercaptoethanol, 0.1 mM PMSF, 50 mM MG132, proteinase inhibitor cocktail), Buffer 2 (0.25 M Sucrose, 10 mM Tris-HCL pH8, 10 mM MgCl$_2$, 1% Triton

X-100, 5 mM ß-mercaptoethanol, 0.1 mM PMSF, 50 mM MG132, proteinase inhibitor cocktail), and Buffer 3 (1.7 M Sucrose, 10 mM Tris-HCL pH8, 0.15% Triton X-100, 2 mM MgCl$_2$, 5 mM ß-mercaptoethanol, 0.1 mM PMSF, 50 mM MG132, proteinase inhibitor cocktail). Nuclei were then resuspended in nuclei lysis buffer (50 mM Tris-HCL pH8, 10 mM EDTA, 1% SDS, 50 mM MG132, proteinase inhibitor cocktail), sonicated for 10 × 30 s, and diluted 10X in Dilution Buffer (0.01% SDS, 1% Triton X-100, 1.2 mM EDTA, 16.7 mM Tris-HCL pH8, 167 mM NaCl). Next, incubation with the corresponding antibody (anti-GFP (Invitrogen Cat# A11122) at 1:500 dilution, or anti-HA (Abcam Cat# 9110) at 1:100 dilution was performed overnight at 4 °C, and immunoprecipitation was performed using dynabeads. Washes were done sequentially in Low Salt Buffer (0.1% SDS, 1% Triton X-100, 2 mM EDTA, 20 mM Tris-HCL pH 8, 150 mM NaCl), High Salt Buffer (0.1% SDS, 1% Triton X-100, 2 mM EDTA, 20 mM Tris-HCL pH 8, 500 mM NaCl), LiCl Buffer (0.25 M LiCl, 1% NP40, 1% deoxycholic acid sodium, 1 mM EDTA, 10 mM Tris-HCL pH 8), and TE 1X. Immunocomplexes were eluted in Elution Buffer (1% SDS, 0.1M NaHCO$_3$), de-crosslinked in 10 mM NaCl overnight at 65 °C, and treated with proteinase K. DNA was then column purified, eluted in 100 µL of elution buffer, and 2 µl were used for qPCR (ChIP-qPCR) using *KAT1* promoter-specific primers (Supplementary Table 2) spanning the region containing the predicted binding sites for the PIFs and an intergenic region as a negative control[49]. Three independent biological replicates were performed and PIF binding was represented as % of input and relative to Col-0 set at unity.

### Yeast one hybrid assay

Y1H assays were performed following previously described protocols[81]. Bait DNA regions encompassing region P1 or P2 in the promoter of *KAT1* (Fig. 5a) were cloned in the pTUY1H vector, containing the wild-type G-box and PBE-box sequences (CACGTG and CACATG, respectively) (WT), or the mutated versions CCCGTG and CCCATG (Mut) previously shown to impair PIF binding[82]. Cloning was performed by annealing of complementary oligonucleotides (Supplementary Table 3) followed by XmaI-XbaI restriction. PIF3 was cloned in the pDEST22 vector as a hybrid prey protein GAL4-Activation Domain (GAD)-PIF3. pTUY1H and pDEST22 confer yeast selection in leucine (L) and tryptophan (W), respectively, and pTUY1H carries the reporter gene *HIS3* which allows growth in the absence of histidine (H). Transformation was performed by heat shock following standard procedures (Clontech) to introduce the cloned GAD-PIF3 and the P1 and P2 DNA baits into *S. cerevisiae* pJ694a (a mating type) and Y187 (α mating type), respectively, and transformants were selected in dropout base medium (DOB)-W (for GAD-PIF3) or (DOB)-L (for DNA baits) plates. Both strains were combined by mating in 1.5 mL tubes, and after diploid enrichment by growth in DOB-L-W media, cells were plated in -L-W to assess growth and -L-W-H to detect interactions, using 3-amino-1,2,4-triazole (3-AT) to block bait activation of the *HIS3* reporter. We observed the difference in growth after 3-4 days at 28 °C.

### Diurnal profile expression

Transcript abundance were analyzed using the publicly available genome-wide expression data DIURNAL5 (http://diurnal.mocklerlab.org)[83] using a cut-off of 0.2 for the following conditions: SD (Col-0_SD) names as Col-0 SD and free running (LL23_LDHH) named as Col-0 LL.

### Reporting summary

Further information on research design is available in the Nature Portfolio Reporting Summary linked to this article.

## Data availability

The authors declare that the data supporting the findings of this study are available within the paper and its supplementary information files. Publicly available datasets can be accessed at http://diurnal.mocklerlab.org, https://www.frontiersin.org/articles/10.3389/fpls. 2016.00962/full, and https://bmcgenomics.biomedcentral.com/articles/10.1186/1471-2164-12-216#Sec25. Source data are provided with this paper.

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

## Acknowledgements
We are grateful to ABRC/NASC for *kat1* and *aba2* seeds, and to Luis Oñate-Sánchez for Y1H materials, protocols, and advice. This work was supported by grants from FEDER/Ministerio de Ciencia, Innovación y Universidades – Agencia Estatal de Investigación (Project References BIO2015-68460-P, PGC2018-099987-B-I00, and PID2021-122288NB-I00 to E.M.; PID2019-10454GB-I00 to L.Y.), from the CERCA Programme/ Generalitat de Catalunya (Project Reference 2017SGR-718) to E.M, and from the Daiwa Anglo Japanese Foundation (1019/13721) and the Royal Society (IEC\R1\180047) to G.T-O. We acknowledge financial support from the Spanish Ministry of Economy and Competitiveness, through the 'Severo Ochoa Programme for Centres of Excellence in R&D' 2016–2019 (SEV-2015-0533) and CEX2019-000902-S funded by MCIN/AEI/ 10.13039/501100011033. A.B.-M. was supported by the predoctoral program AGAUR-FI ajuts (2023 FI-1 00910) Joan Oró of the Secretariat of Universities and Research of the Department of Research and Universities of the Generalitat of Catalonia and the European Social Plus Fund, and M.Q. received postdoctoral funding from the European Union's Horizon 2020 research and innovation programme under the Marie Skłodowska-Curie grant agreement no. 945043.

## Author contributions
E.M., P.L., A.R., and N.V. conceived the project and planned the experiments. A.R., A.B.-M., M.Q., G.T.-O., and N.V. performed experiments and analyzed the data. A.L. and L.Y. provided the KAT1-OX line. E.M., P.L., A.R., and N.V. wrote the manuscript. All authors discussed and commented on the manuscript.

## Competing interests
The authors declare no competing interests.
