## [Peer Review File · Nature Communications]

PIF transcriptional regulators are required for rhythmic stomatal movementsReviewer #1 (Remarks to the Author):

This study reports a role of PIFs (particularly PIF3) in blue-light induced stomata opening following the previous night period. At the mechanistic level, the authors demonstrated that PIFs accumulation in the previous dark period is required for promoting KAT1 expression through direct binding to its promoter. Light induces degradation of PIF proteins and thus stomata close at the end of the day under SD conditions. They also showed that endogenous basal level of ABA is required to maintain stomata closure at the night period. The data presented, for most parts, is solid and conclusions are well justified. Overall, this study provides significant new insights into the role of PIFs and regulation of stomata opening by diurnal light/dark cycles.

I have several comments for the authors to address:

1. Please explicitly explain how stomata opening was measured under the dark period. (green safelight? Is it really safe? Is it sufficient to observe the stomata aperture under such dim light?)
2. Fig2b, first panel, in *pif3*, what is the green signal?
3. The authors identified KAT1 as a putative direct target gene of PIFs by looking for genes that are inversely regulated by PIFs and ABA. The data of Wang et al. (20, BMC Genomics, 12, 216) were used. Could the authors explain more explicitly how these genes were identified as GC-specific genes?
4. It might be worthwhile to specify the light regimes used for each experiment, particularly the data depicted in each data Figure (light spectrum and intensities, given the differential roles of red and blue light on stomata movement).
5. The authors claimed that KAT1 is a direct target gene of PIFs based on ChIP-PCR assay. This conclusion needs to be substantiated with additional experiments (such as Yeast one-hybrid assay and EMSA, with mutagenesis of the suspected binding site).
6. The model presented in 7b might be simplified by removing the dark regulation, focusing on the antagonistic role of red and blue light?

Reviewer #2 (Remarks to the Author):

Summary: The opening and closing of stomata – small epidermal pores necessary for gas exchange – is a vital process for plant fitness and productivity and is thus tightly regulated in response to environmental factors such as light, temperature and humidity. In their manuscript, Ravira et al. link PIF transcription factors, key regulators of the light signalling pathway, to diurnal patterns in stomatal aperture. Specifically, they observe that the light-induced opening of stomata after dawn is abolished in a *pif* quadruple (*pifQ*) mutant. This phenotype is dependent on the preceding dark phase, in which PIF proteins accumulate. The authors link the *pifQ* phenotype to reduced levels of the KAT1 transcript, which encodes a stomata-specific potassium ion channel, and *kat1* mutants indeed do not display stomatal opening after dawn. The authors suggest a model in which PIF proteins induce KAT1 expression at the end of the night, and high KAT1 levels in the morning subsequently promote a turgor increase in guard cells that results in stomatal opening.

Significance: The regulation of stomatal opening is directly linked to a plant's water-use efficiency and drought responses, aspects that are highly relevant in the context of climate change. Blue light-induced stomatal opening is largely attributed to phototropin photoreceptors (Matthews et al., 2020, J Exp Bot). However, other light signalling components have been linked to the regulation of stomatal aperture: The E3 ligase COP1 promotes dark-induced stomatal closure downstream of cryptochrome and potentially phototropin blue-light receptors (Mao et al., 2005, PNAS). PIFs themselves have been shown to affect stomatal aperture in rice and maize, coordinating red light and ABA responses (Li et al., 2022, The Plant Cell). It is thus not surprising that Arabidopsis PIFs affect this process as well; the main advancement of the manuscript is to link PIF function to diurnal patterns in stomatal aperture, which is of interest mainly for a very specialist audience.

Soundness of the research: The experimental design and data analysis throughout the manuscript are sound, but some aspects (especially the role of KAT1 in PIF-dependent stomatal opening) need to be further corroborated.

Major points:

1) The authors convincingly lay out that KAT1 is directly regulated by PIFs. I am however not convinced that KAT1 expression is indeed the major factor controlling PIF-dependent stomatal opening in the morning given that it is upregulated just 2.3-fold (compared to a 24-fold downregulation in response to ABA). PIFs regulate expression of hundreds of genes, and the key downstream genes may not necessarily ABA-responsive. To further corroborate an essential function of KAT1 downstream of PIFs, additional epistasis analyses would be required:

- Can KAT1 overexpression in the *pifQ* background restore stomatal opening?
- Does a PIF overexpressor line display altered stomatal aperture (e.g. increased or extended stomatal opening) and if so, is this effect abolished in a *kat1* mutant?
- Is increased KAT1 expression at the end of the night reflected at the protein level (e.g. based on a translational fusion line)? See also point 2.

2) The authors observe that PIF-dependent stomatal opening is not triggered by red light, but still occurs in response to blue light after a 3 h red light treatment (Fig. 3). Do PIF protein levels persist during red light treatment? How does this reflect on KAT1 transcript and protein levels (i.e. do they persist during the red light treatment as well)?

Minor points

3) PIF appear to be involved specifically in blue light-dependent stomatal opening. Is this downstream of phototropin or cryptochrome photoreceptors, or both?

Reviewer #3 (Remarks to the Author):

The article found that PIF affects rhythmic stomata movements in daily dark/light cycles by regulating K⁺ channel gene KAT1.

- Can the sentences like "suppressing ABI activity and leading to the activation of OST1, CPKs and GHR1" use ABI directly, referring to ABI1 and ABI2?
- Abbreviations appearing for the first time should be written in full, and can then be used. Check the manuscript and revise them.
- "and is thought to coordinate stomatal opening with photosynthesis in the mesophyll cells", in the mesophyll cells and guard cells (Matthews et al. 2020).
- In Fig.1
 - The legend showed that stomata aperture expressed as area in (a) and as stomata width in (b), whether the ordinate in Fig.1a and so on using stomata aperture is accurate.
 - How do the stomatal movements of the 3 lines change in the daily dark/light cycles of normal grown environment and whether they have been observed?
 - Under SD conditions, humidity and temperature are constant, why the area and width of stomata drop sharply when ZT3-ZT6.
 - Whether Fig.1d, Fig.1a, Fig.1b, SI Fig. 2, and SI Fig. 3 were the same batch of materials and measured at the same time? Why is the degree of change different in different figures under the same conditions. For example, in Fig.1a and Fig.1d, ZT0-ZT3 stomatal area under SD conditions showed great differences. Why is there no difference in stomatal area between Col-0 and *aba2* in ZT3 in Fig.1a, while there is a difference in width in Fig.1b? ZT24 is the opposite. Is there also a difference in stomatal length? Does area refer to stomatal pore area? What is the statistical standard for area? Is there a difference in stomatal density?
 - Why did the *pifq* stomata still respond at ZT3 under SD conditions? The conclusion "PIFs are required for morning stomata opening" lacks evidence to support it.
- How do "the results" lead to this conclusion "The results above prompted us to hypothesize that PIFs, which accumulate during the night in SD, and are necessary under these conditions for dynamic responses such as hypocotyl elongation, might be involved in stomata movements". What is "these conditions"? Sentences should be clear, easy to understand, and logical.
- What exactly is *pifq*, which is best explained in the article. *pifq* includes 4 genes, which PIF gene is necessary, also need to be verified and explained.

7. "under LL conditions, where PIF3 and likely other PIFs do not accumulate", Why mention PIF3.
8. PIF affects stomata opening at dawn. Will the degree of stomata opening be greater when PIF is overexpressed or will it be similar to Col-0 due to homeostasis? PIF accumulates at night and degrades during the day. Previous studies have found that PIF promotes stomata closure. If the conclusion of this article is correct, does it conflict with previous researches?
9. "Remarkably, in sharp contrast to the WT, *aba2* stomata then opened progressively during the night to display fully open stomata at ZT24 (~34 μm^2) (Fig. 1a-c). These results suggest that the main role of basal ABA under SD conditions is to maintain the stomata closed during the night hours", whether the results of endogenous ABA content changes would be more helpful to understand, if possible. Endogenous basal ABA prevents early stomata opening during the night, what about ABA deficiency? What about ABA and PIF accumulate or lack at the same time, and what happen when one accumulates and one lacks?
10. Why only focus on PIF3? PIF3 is localized in the cell nucleus and is degraded by light, which has been demonstrated. What is the significance of this experiment. Would it be more straightforward to test for PIF protein levels at different times.
11. In Fig.3a, there is little difference in stomata opening between Col-0 and *pifq* at ZT6. If the observation time is extended, will it be consistent? Why? In the title "PIFs are necessary for blue-light induced stomatal opening", rapid blue-light induced stomatal opening may be more appropriate. In Fig.3b, why do visible stomata phenotypes not follow the time point in Fig.3a. The legend said that "PIFs promote blue-light induce stomata opening in the morning of dark/light cycles", dark/light cycles were not reflected in the experiment.
12. "Identification of the inward-rectifying potassium channel KAT1 as a PIF-induced and ABA-repressed guard cell-specific gene under SD conditions", what about the normal grown environment?
13. "To explore the possibility that PIFs regulate expression of a necessary component for stomatal opening in the phototropin-mediated pathway, and given the co-regulation of stomata movements by PIFs and ABA, we next aimed to identify ABA-responsive genes in guard cells that might be PIF targets", there was no experiment on the co-regulation of stomatal movement by PIF and ABA. How to draw the conclusion?
14. "To this end, we compared previously defined gene sets of PIF-regulated genes in SD at the end of the night (538 genes, 331 induced and 207 repressed by PIFs) with ABA-responsive guard-cell specific genes (906 genes, 515 induced and 384 repressed by ABA) (responsive to a treatment of 50 μM of exogenous ABA for 3 hours)", what were the samples in the experiment? Were the genes found that were known to be regulated by ABA or PIF?
15. "Interestingly, one of these genes (AT5G46240) encodes the KAT1 voltage-dependent potassium channel predominantly expressed in GC (Fig. 4b)", AT5G46240 is KAT1? Pay attention to the accuracy of sentence description.
16. KAT1 is regulated by ABA and PIF, and whether qPCR or other assays have been conducted by yourselves. There is insufficient evidence that PIFs regulates KAT1. Is KAT1 expression unaffected by light in *pifq*? Do the results in Fig.4b and 4c come from the websites? Have they been repeatedly verified by yourselves? In Fig.4d, why is there only a difference in 0 SD, and no significant difference between *aba2* and Col-0? Would it be better to use KAT1 protein level changes, or add ABA or light treatment?
17. "These data suggest that endogenous basal ABA does not significantly repress PIF-mediated induction of KAT1 expression at night, and might contribute to the repression of KAT1 expression in SD upon exposure to light", how to draw the conclusion?
18. "Taken together, we conclude that in SD PIFs are necessary to induce the expression of KAT1, a GC specific gene that encodes the inward-rectifying K⁺ channel driving the K⁺ uptake that leads to stomata opening upon activation by blue light", would it be better to add red-light and blue-light treatment?
19. ChIP-seq data showed binding of PIF1, PIF3 and PIF4 to the region, why chose PIF3 but not PIF1 or PIF4? Moreover, only a CHIP-qPCR experiment is not enough to convince that KAT1 is directly downstream of PIF. For CHIP-qPCR, primer combinations should be better selected to identify binding sites.
20. Was the K⁺ changes measured in *kat1* and KAT1 OX? Are stomatal movements induced by K⁺ changes in guard cells? Just observing stomatal aperture is not enough to support your claim.
21. Why discuss the similarity of PIF in regulating stomata movement and hypocotyl elongation? Whether it is more appropriate to discuss PIF and circadian rhythm?
22. "Interestingly, recent studies have reported a role for PIFs as negative regulators of stomata

opening under red light. Although we did not detect this effect in our experiments in red light (Fig. 3)", you did not detect this effect because of PIF knockout was used. PIF was degraded under red light, so there would be no difference between Col-0 and pifq, which does not mean that this phenomenon does not exist.

23. Why do you draw three models, including Fig. 7 and SI Fig. 5? Sum them up on a figure, if possible.

24. The overall association of ABA was not strong, and there was insufficient evidence for ABA involved in the regulation of this pathway.

25. At night, PIF accumulation promotes KAT1 transcription, while endogenous ABA inhibits KAT1 transcription. According to this statement, does KAT1 channel increase every day. Whether there is another mechanism to achieve the dynamic balance of KAT1.

26. How cotyledons were sampled to keep stomata constant, how many biological replicates were carried out. Check the full text, all experiments should be carried out in three or more biological replicates.

Reviewer #4 (Remarks to the Author):

In the present manuscript, Rovira and colleagues provide an interesting new look at possible mechanisms that regulate rhythmic stomatal movements during the night-to-day transition. The authors provide evidence that PIF transcription factors are required for stomatal opening in the morning as well as blue light-induced stomatal opening. They also show that PIFs are required for accumulation of guard cell-specific K⁺ channel KAT1 at night. Furthermore, the authors provide evidence that PIF3 binds to the KAT1 promoter by ChIP-qPCR analysis. From these results, the authors conclude that PIF accumulation at night promotes KAT1 expression, which ensures stomatal opening in the morning.

Major concerns

1) Genetic interaction of PIF and KAT1 expression. The authors provided biochemical evidence that PIF3 directly binds to the KAT1 promoter by ChIP-qPCR (Fig. 5b). If the inhibition of stomatal opening seen in the pifq mutant is due to decreased expression of KAT1, then pifq should exhibit stomatal opening by introduction of KAT1OX. Otherwise, the entire argument for PIF-induced KAT1 expression and stomatal opening are flawed. These data would strengthen their conclusion.

2) The authors showed that KAT1OX in Col-0 exhibited larger stomatal opening at ZT0 and ZT3 and slower stomatal closure at ZT6 compared to Col-0 (Fig. 6b). How do they explain stomatal closure in KAT1OX at ZT6? Given that KAT1 is constitutively expressed in KAT1OX, their stomata should open during light irradiation.

3) Furthermore, if the lack of stomatal opening in Col-0 in LL conditions is attributed to decreased expression of KAT1, does KAT1OX show stomatal opening in LL?

4) Previous studies have shown that phytochromes positively regulate light-induced stomatal opening (Wang et al. 2010, Mol Plant, etc). On the other hand, according to the model proposed by the authors (Fig. 7b), phytochromes negatively regulate KAT1 expression and stomatal opening via PIF degradation. The authors need to assay the expression of KAT1 (e.g. Fig. 4d) and stomatal opening (e.g. Fig. 1d) in phytochrome mutants. These data would expand our understanding of phytochrome-mediated control of stomatal movements.

Minor concerns

1) In association with Major concern 2, I recommend the authors to provide KAT1 expression in KAT1OX under stomatal experimental conditions.

2) Fig. 1a, 1b, 1d, 4, 6a, and 6b – please include light/dark bar.

3) No information is provided about any differences in stomatal size among Col-0, aba2, pifq, kat1, and KAT1OX. Since the authors reported stomatal pore area and size, this information is needed to validate the data.

4) Page 4, GHR1 has been reported to be kinase-dead (Sierla et al. 2018). It is not appropriate to quote in this context.

5) Page 4 - OST2 - AHA1/OST2.

6) Page 4 -Na⁺/H⁺ - superscript.

We thank the reviewers for their insightful comments, which we believe have allowed us to greatly improve our previous manuscript by adding new genetic and expression data. The revised paper has been adjusted accordingly, and the changes and additions are in blue color to facilitate identification.

Below are the comments received followed by our answers:

Reviewer #1 (Remarks to the Author):

This study reports a role of PIFs (particularly PIF3) in blue-light induced stomata opening following the previous night period. At the mechanistic level, the authors demonstrated that PIFs accumulation in the previous dark period is required for promoting KAT1 expression through direct binding to its promoter. Light induces degradation of PIF proteins and thus stomata close at the end of the day under SD conditions. They also showed that endogenous basal level of ABA is required to maintain stomata closure at the night period. The data presented, for most parts, is solid and conclusions are well justified. Overall, this study provides significant new insights into the role of PIFs and regulation of stomata opening by diurnal light/dark cycles.

We thank this Reviewer for the positive and supportive summary of the contributions of the manuscript.

I have several comments for the authors to address:

1. Please explicitly explain how stomata opening was measured under the dark period. (green safelight? Is it really safe? Is it sufficient to observe the stomata aperture under such dim light?)

We apologize for not giving sufficient detail. In the dark period, cotyledons were wet mounted in the dark with a dim green safelight. Then, images were taken with a Zeiss AxioPhot DP70 optical microscope using the same settings we used for the light samples. Illumination with the microscope light allowed observation and photography of the stomata in the mounted cotyledons. The imaging process was fast, which we deemed appropriate to keep the dark state of the stomata, as evidenced by the observed differences in stomata opening with the samples taken during the light time points.

We have now added more detail in the Methods section (line 448).

2. Fig2b, first panel, in pif3, what is the green signal?

The signal corresponds to auto-fluorescence of the guard cells, which is always seen around the pore. We have now added this remark in Fig. 2b legend (line 545).

3. The authors identified KAT1 as a putative direct target gene of PIFs by looking for genes that are inversely regulated by PIFs and ABA. The data of Wang et al. (20, BMC Genomics, 12, 216) were used. Could the authors explain more explicitly how these genes were identified as GC-specific genes?

Wang et al. employed a novel protocol to isolate epidermal peels with guard cells as the only intact cell type. These epidermal peels were then used as the source of guard cell RNA for their microarray analysis comparing ABA-treated with solvent-treated, to identify ABA-responsive genes in GC. Next, they compared this gene set to ABA-responsive genes in

whole leaves, to define a set of genes that are ABA-responsive and guard-cell specific (not detected in whole leaves) (906 genes, 515 induced and 384 repressed by ABA). We have revised our text and added more detail in this part to improve clarity (line 235).

4. It might be worthwhile to specify the light regimes used for each experiment, particularly the data depicted in each data Figure (light spectrum and intensities, given the differential roles of red and blue light on stomata movement).

Thanks for the suggestion. We are now specifying in the Methods section the light intensities in our SD and LL conditions, as well as the red and blue light ratio of our white light source (lines 434-444). In addition, we are now showing the light spectra of our white, red and blue light sources in the new SI Fig.13.

5. The authors claimed that KAT1 is a direct target gene of PIFs based on ChIP-PCR assay. This conclusion needs to be substantiated with additional experiments (such as Yeast one-hybrid assay and EMSA, with mutagenesis of the suspected binding site).

We have now performed a yeast one-hybrid assay with regions P1 and P2 from the KAT1 promoter. Following also the suggestion by Reviewer 3, we are now showing specific binding of PIF3 to the P2 region by yeast one hybrid (new SI Fig. 8). Compared to the plasmid expressing GAD alone, GAD-PIF3 was able to enhance binding to the P2 region in the KAT1 promoter. Importantly, binding was lost in a mutated P2 region harboring a mutated version of the G-box motif, supporting direct binding to the KAT1 promoter through the G-box motif in the P2 region. We could not detect specific binding of GAD-PIF3 to the P1 region, in accordance to our ChIP-qPCR data in Fig. 3b.

We have updated the Methods section to include the yeast one-hybrid assay (lines 505-520). Results are shown in new Fig. S8 and discussed in the main text (lines 286-287).

6. The model presented in 7b might be simplified by removing the dark regulation, focusing on the antagonistic role of red and blue light?

We have followed the suggestion and simplified the model in 7b by removing the dark regulation.

Reviewer #2 (Remarks to the Author):

Summary: The opening and closing of stomata – small epidermal pores necessary for gas exchange – is a vital process for plant fitness and productivity and is thus tightly regulated in response to environmental factors such as light, temperature and humidity. In their manuscript, Ravira et al. link PIF transcription factors, key regulators of the light signalling pathway, to diurnal patterns in stomatal aperture. Specifically, they observe that the light-induced opening of stomata after dawn is abolished in a pif quadruple (pifQ) mutant. This phenotype is dependent on the preceding dark phase, in which PIF proteins accumulate. The authors link the pifQ phenotype to reduced levels of the KAT1 transcript, which encodes a stomata-specific potassium ion channel, and kat1 mutants indeed do not display stomatal opening after dawn. The authors suggest a model in which PIF proteins induce KAT1 expression at the end of the night, and high KAT1 levels in the morning subsequently promote a turgor increase in guard cells that results in stomatal opening.

Significance: The regulation of stomatal opening is directly linked to a plant's water-use efficiency and drought responses, aspects that are highly relevant in the context of climate change. Blue light-induced stomatal opening is largely attributed to phototropin photoreceptors (Matthews et al., 2020, J Exp Bot). However, other light signalling components have been linked to the regulation of stomatal aperture: The E3 ligase COP1 promotes dark-induced stomatal closure downstream of cryptochrome and potentially phototropin blue-light receptors (Mao et al., 2005, PNAS). PIFs themselves have been shown to affect stomatal aperture in rice and maize, coordinating red light and ABA responses (Li et al., 2022, The Plant Cell). It is thus not surprising that Arabidopsis PIFs affect this process as well; the main advancement of the manuscript is to link PIF function to diurnal patterns in stomatal aperture, which is of interest mainly for a very specialist audience.

Soundness of the research: The experimental design and data analysis throughout the manuscript are sound, but some aspects (especially the role of KAT1 in PIF-dependent stomatal opening) need to be further corroborated.

We thank this Reviewer for the positive and supportive summary of the contributions of the manuscript.

Major points:

1) The authors convincingly lay out that KAT1 is directly regulated by PIFs.

I am however not convinced that KAT1 expression is indeed the major factor controlling PIF-dependent stomatal opening in the morning given that it is upregulated just 2.3-fold (compared to a 24-fold downregulation in response to ABA). PIFs regulate expression of hundreds of genes, and the key downstream genes may not necessarily ABA-responsive. To further corroborate an essential function of KAT1 downstream of PIFs, additional epistasis analyses would be required:

a) Can KAT1 overexpression in the *pifQ* background restore stomatal opening?

Thank you for this important suggestion. We agree that additional genetic evidence was necessary to fully support the proposed pathway. We have therefore generated *pifQKAT1OX* mutants as suggested by this reviewer to test their genetic interaction. As shown in new Fig.6c and new SI Fig. 10, KAT1 overexpression was able to completely restore the stomatal opening phenotype in *pifQ*, both under SD conditions (new Fig. 6c) and in response to blue light (new SI Fig. 10). These results support the essential function of KAT1 downstream of PIFs in the regulation of stomata opening.

These new important data are discussed in the text in lines 308-313.

b) Does a PIF overexpressor line display altered stomatal aperture (e.g. increased or extended stomatal opening) and if so, is this effect abolished in a *kat1* mutant?

We have tested the stomata aperture in PIF3-OX (and also in PIF5-OX) and have not detected any significant differences in stomatal opening. *KAT1* expression was also not significantly altered in PIFOX. These results suggest tight endogenous regulation of *KAT1* mRNA expression and/or stability.

This new data are included in SI Fig. 4b, 4c and discussed in lines 164-165, and 256-258.

c) Is increased KAT1 expression at the end of the night reflected at the protein level (e.g. based on a translational fusion line)? See also point 2.

We did not attempt to detect KAT1 protein levels for this current work. KAT1 protein has only been detected when overexpressed either transiently (Sutter et al., Plant Cell 2006; Curr Biol 2007; Sieben et al., Plant Journal 2008; Lebaudy et al., JBC 2009), or stably (Yenush laboratory, unpublished). In addition, according to previous literature, ABA is able to induce endocytosis of KAT1 (Sutter et al., Curr Biol 2007), which adds an additional layer of protein regulation. Although of great interest, we feel that detailed dynamics of the KAT1 protein are beyond the scope of this work.

2) The authors observe that PIF-dependent stomatal opening is not triggered by red light, but still occurs in response to blue light after a 3 h red light treatment (Fig. 3). Do PIF protein levels persist during red light treatment?

Under the same SD conditions, we previously reported progressive accumulation of PIF3 protein during the night hours, and fast (within 1 h) phytochrome-mediated red-light induced PIF3 degradation (Soy et al., Plant Journal 2012). This light-induced degradation is slower for other PIFs such as PIF4, which can still be detected during the first hours of light (Bernardo-García et al., Genes&Devel 2014).

How does this reflect on KAT1 transcript and protein levels (i.e. do they persist during the red light treatment as well)?

We tested *KAT1* transcript levels in Col-0 after 3h of red light (ZT3). We found that there is a downregulation with respect to the dark point at ZT0 of approximately 30% (new SI Fig. 6). Although we do not know how this reflects on KAT1 protein levels, the finding that overexpression of KAT1 is able to complement the *pifq* phenotype and fully restore *pifq* stomatal opening in blue light after the 3h red light treatment (new SI Fig. 10), strongly suggests that (a) it is the lack of KAT1 in *pifq* that is impairing stomatal opening, and (b) in Col-0, KAT1 protein persists (or it is being produced) under red light and it is available upon treatment with blue light to allow stomata opening.

Minor points

3) PIF appear to be involved specifically in blue light-dependent stomatal opening. Is this downstream of phototropin or cryptochrome photoreceptors, or both?

Blue-light dependent stomatal opening is mediated by phototropins (Boccalandro et al., Plant Physiol 2012). Although a previous involvement of cryptochrome was reported (Mao et al., PNAS 2005), it was later found that the effects of cry on stomatal conductance were largely indirect and involved the control of ABA levels (Boccalandro et al., Plant Physiol 2012).

The regulation of stomatal opening reported here represents an interesting photoreceptor crosstalk between the phototropins and the phytochromes (and to lesser extend the cryptochrome), responsible for the rhythmic accumulation of the PIFs to allow maximum PIF accumulation and activity at the end of the night to induce KAT1 expression.

Reviewer #3 (Remarks to the Author):

The article found that PIF affects rhythmic stomata movements in daily dark/light cycles by regulating K⁺ channel gene KAT1.

1. Can the sentences like “suppressing ABI activity and leading to the activation of OST1, CPKs and GHR1” use ABI directly, referring to ABI1 and ABI2?

We would like to thank this Referee for the thorough detail of the comments. The suggested change has now been made (line 71).

2. Abbreviations appearing for the first time should be written in full, and can then be used. Check the manuscript and revise them.

The manuscript has now been revised to include all the names written in full on first appearance.

3. “and is thought to coordinate stomatal opening with photosynthesis in the mesophyll cells”, in the mesophyll cells and guard cells (Matthews et al. 2020).

Thanks for noticing this incorrection. The suggested change has been made (line 90).

4. In Fig.1

1) The legend showed that stomata aperture expressed as area in (a) and as stomata width in (b), whether the ordinate in Fig.1a and so on using stomata aperture is accurate.

We have revised all the graphs and the ordinates are accurate. In accordance to other authors, we have chosen to express stomata opening mainly as pore area (as in Fig. 1a), although we have included width (Fig. 1b), and we are now also including stomata pore length (SI Fig. 2) to better capture the nature of the stomata dynamics in the mutant background compared to Col-0.

2) How do the stomatal movements of the 3 lines change in the daily dark/light cycles of normal grown environment and whether they have been observed?

We chose to focus our experimental design on a situation where humidity and temperature were constant. Our choice of SD photoperiod and comparison with LL and DD were based on previous studies on the rhythmic response of hypocotyl elongation (Nozue et al., Nature 2007; Soy et al., Plant J 2012). We feel that this strategy has successfully allowed us to identify the PIFs as major regulators of stomata dynamics, and their interplay with basal ABA levels.

3) Under SD conditions, humidity and temperature are constant, why the area and width of stomata drop sharply when ZT3-ZT6.

As pointed out, humidity and temperature are maintained constant in our experimental setup. However, other cues such as time of day are not, which can affect fluctuations for example of hormone levels. In fact, this sharp drop between ZT3-ZT9 is slower in *aba2* (Fig.1a), suggesting a role for endogenous ABA in the regulation as discussed in lines 171-173 and 179-180. However, because closure does take place in *aba2*, we propose that additional factor(s) are probably involved (line 180).

4) Whether Fig.1d, Fig.1a, Fig.1b, SI Fig. 2, and SI Fig. 3 were the same batch of materials and measured at the same time? Why is the degree of change different in different figures

under the same conditions. For example, in Fig.1a and Fig.1d, ZT0-ZT3 stomatal area under SD conditions showed great differences.

We have used different batches of materials (each batch always including the appropriate controls and mutant genotypes) grown at different times. Seeds from different batches were never mixed. Although the degree of change in different figures might vary slightly, we feel that the slight variation probably reflects the variation in biological material.

Importantly however, the main conclusions are always supported. For example in Fig.1a and Fig.1d, *pifq* in SD is always defective in stomatal opening compared to Col-0.

Why is there no difference in stomatal area between Col-0 and *aba2* in ZT3 in Fig.1a, while there is a difference in width in Fig.1b? ZT24 is the opposite.

Is there also a difference in stomatal length?

We are now showing stomata length in a time course analysis (presented in new SI Fig 2) . Indeed, stomata length in *aba2* shows variations that can account for the differences between area and width (line 174).

Does area refer to stomatal pore area? What is the statistical standard for area? Is there a difference in stomatal density?

We apologize for not having made this clear. Stomatal area indeed refers to stomatal pore area, and measurements represent mean values (n=40-100) \pm standard error (SE). We have revised the text and figure legends accordingly. Regarding stomata density, preliminary results did not find significant differences between Col-0 and *aba2*, which would reflect deficiencies in stomata development.

5) Why did the *pifq* stomata still respond at ZT3 under SD conditions? The conclusion “PIFs are required for morning stomata opening” lacks evidence to support it.

Whereas WT stomata pore area was in average $\sim 18 \mu\text{m}^2$ at ZT0, and opened to reach an aperture of $\sim 37 \mu\text{m}^2$ between 0-3 hours after dawn (ZT0-ZT3), *pifq* stomata displayed similarly closed stomata at ZT0 but only reached an aperture of $\sim 23 \mu\text{m}^2$ at ZT3 (Fig 1a). Similar results were observed when stomata width was measured (Fig. 1b). We therefore believe that our conclusion “PIFs are required for morning stomata opening” is strongly supported by our genetic data.

The slight response of *pifq* at ZT3 might reflect the role of other PIFs in inducing stomata opening (PIF6 and/or PIF7), paralleling that of hypocotyl elongation under SD. Indeed, regulation of PIF-mediated hypocotyl elongation is severely affected in *pifq*, but it is only completely abolished in *pifqpif7* (Leivar et al., 2020).

5. How do “the results” lead to this conclusion “The results above prompted us to hypothesize that PIFs, which accumulate during the night in SD, and are necessary under these conditions for dynamic responses such as hypocotyl elongation, might be involved in stomata movements”. What is “these conditions”? Sentences should be clear, easy to understand, and logical.

We have now reworded the sentence to improve clarity. The sentence now reads:

“The results in the previous section prompted us to hypothesize that PIFs, which accumulate during the night in SD, and are necessary under these SD conditions for dynamic responses such as hypocotyl elongation, might be involved in stomata movements”. Lines 143-145.

6. What exactly is *pifq*, which is best explained in the article. *pifq* includes 4 genes, which PIF gene is necessary, also need to be verified and explained.

We apologize for the confusion with the mutant name. *pifq* is the *pif quadruple* mutant deficient in 4 PIFs (PIF1, PIF3, PIF4, PIF5). Now this is clarified in the main text at first mention (line 146).

Following your suggestion, we have tested stomata opening dynamics in the *pif* single mutants *pif1*, *pif3*, *pif4*, and *pif5*. We have not found significant differences in stomata aperture at ZT3, suggesting functional redundancy among PIFs in the regulation of stomatal dynamics. These results are shown in the new SI Fig. S4 and discussed in the text (lines 162-164).

7. “under LL conditions, where PIF3 and likely other PIFs do not accumulate”, Why mention PIF3.

Our laboratory has previously shown lack of accumulation of endogenous PIF3 protein under the same LL conditions as the ones used here (Soy et al., Plant J 2012). This has not yet been shown for other PIFs, although given the similar phytochrome-imposed regulation it is likely to be similar.

8. PIF affects stomata opening at dawn. Will the degree of stomata opening be greater when PIF is overexpressed or will it be similar to Col-0 due to homeostasis?

As suggested, we have now checked stomata opening in PIF-OX and found that it is similar to Col-0. We found that *KAT1* expression is also not significantly altered in PIF-OX with respect to Col-0. Both results suggest tight regulation of *KAT1* expression and stomata dynamics in response to increased levels of PIFs, due to homeostasis as indicated by the reviewer. These new data are shown in new SI Fig. 4 and discussed in the text in lines 164-165 and 256-258.

PIF accumulates at night and degrades during the day. Previous studies have found that PIF promotes stomata closure. If the conclusion of this article is correct, does it conflict with previous researches?

Because different conditions were used in previous studies, we believe that our results do not conflict with them and are instead complementary. In the mentioned studies, light conditions were kept constant (as opposed to our alternating day/night conditions) and/or water stress was applied (in contrast to our well-watered conditions) (Wang et al. Mol Plant 2010; Li et al. Plant Cell 2022). We provide discussion of these previous results in contrast to our findings in lines 367-374.

9. “Remarkably, in sharp contrast to the WT, *aba2* stomata then opened progressively during the night to display fully open stomata at ZT24 (~34 μm^2) (Fig. 1a-c). These results suggest that the main role of basal ABA under SD conditions is to maintain the stomata closed during the night hours”, whether the results of endogenous ABA content changes would be more helpful to understand, if possible. Endogenous basal ABA prevents early stomata opening during the night, what about ABA deficiency? What about ABA and PIF accumulate or lack at the same time, and what happen when one accumulates and one lacks?

Our results using the ABA-deficient mutant *aba2/gin1-3* (Cheng et al., Plant Cell 2002) showing open stomata at night in clear contrast to Col-0 (Fig. 1a-c), strongly supports our conclusion that the role of endogenous basal ABA levels (accumulated in Col-0 but not in the *aba2* mutant) have the role to maintain stomata closed during the night hours. We believe our genetic evidence strongly supports our interpretation.

Regarding ABA and PIF accumulation, previous studies have found that ABA accumulates at night and is degraded upon exposure to light, similar to PIF3 (Weatherwax et al., Plant Phys 1996; Assmann and Shimazaki, Plant Phys 1999; Soy et al., Plant J 2012). A situation where PIF protein is accumulated to higher levels is now examined in PIF-OX lines (new SI Fig. S4), where stomata dynamics were not affected significantly. Furthermore, we chose to focus on how basal levels of ABA could affect stomatal dynamics by using the ABA-deficient mutant *aba2*. Finally, the *pifq* mutant represents a situation where PIFs lack while basal ABA accumulates. We believe the combination of these genetic resources have allowed us to successfully identify PIF-KAT1 module as a novel regulator of stomata dynamics.

10. Why only focus on PIF3? PIF3 is localized in the cell nucleus and is degraded by light, which has been demonstrated. What is the significance of this experiment. Would it be more straightforward to test for PIF protein levels at different times.

We presume that this Reviewer is referring to Fig. 2, where we show accumulation of PIF3 in the nucleus of guard cells and its degradation upon exposure to light. Indeed, this has been previously demonstrated in hypocotyl cells and whole seedlings. Showing guard cell specific accumulation of PIF3 was necessary to support a role for PIF3 in the guard cell to mediate stomata opening locally (lines 188-189).

11. In Fig.3a, there is little difference in stomata opening between Col-0 and *pifq* at ZT6. If the observation time is extended, will it be consistent? Why?

We have now extended the experiment to ZT9 (new SI Fig. 10b) and the differences in stomata opening between Col-0 and *pifq* are sustained. Importantly, this difference is complemented by overexpressing KAT1 in the *pifq* background, as *pifqKAT1OX* reaches similar opening to Col-0 in response to blue light (new SI Fig. 10b).

In the title “PIFs are necessary for blue-light induced stomatal opening”, rapid blue-light induced stomatal opening may be more appropriate. In Fig.3b, why do visible stomata phenotypes not follow the time point in Fig.3a. The legend said that “PIFs promote blue-light induce stomata opening in the morning of dark/light cycles”, dark/light cycles were not reflected in the experiment.

Thanks for the suggestion. We have changed the title to rapid blue-light induced stomatal opening as suggested (line 199).

In Fig. 3b, we chose to display representative stomata in red (ZT1, ZT3), after addition of blue light to red (ZT4-ZT6), and a control kept in red light (last ZT6 point). We understand the point made by this Reviewer that the time points do not completely align between 3a and 3b, but we feel that the added colored lines (Fig. 3a) and colored bar (Fig. 3b) help with the interpretation.

Regarding “PIFs promote blue-light induce stomata opening in the morning of dark/light cycles”, seedlings were grown in the dark/light cycles of SD (specified in line 207 and Fig Legend line 548).

12. "Identification of the inward-rectifying potassium channel KAT1 as a PIF-induced and ABA-repressed guard cell-specific gene under SD conditions", what about the normal grown environment?

As explained above, we have chosen to focus on well-watered conditions and SD to focus on the effect of basal ABA (as opposed to increased or exogenously applied ABA), and in conditions where PIF accumulation is at a maximum given the long night (16h). When light conditions are varied, for example in continuous light (LL) or continuous darkness (DD), seedlings did not open their stomata in the subjective morning (Fig 1d, SI Fig. 3).

Together, our data indicate that under diurnal conditions morning stomata opening requires (1) PIF accumulation during the previous night, and (2) the transition to light in the morning. This indicates that darkness during the night period (to allow for PIF accumulation) and the transition to light at dawn, are both necessary for stomata movements under diurnal conditions.

13. "To explore the possibility that PIFs regulate expression of a necessary component for stomatal opening in the phototropin-mediated pathway, and given the co-regulation of stomata movements by PIFs and ABA, we next aimed to identify ABA-responsive genes in guard cells that might be PIF targets", there was no experiment on the co-regulation of stomatal movement by PIF and ABA. How to draw the conclusion?

We agreed with this comment and changed it to "regulation of stomata movements by PIFs and ABA" (line 227).

14. "To this end, we compared previously defined gene sets of PIF-regulated genes in SD at the end of the night (538 genes, 331 induced and 207 repressed by PIFs) with ABA-responsive guard-cell specific genes (906 genes, 515 induced and 384 repressed by ABA) (responsive to a treatment of 50 μ M of exogenous ABA for 3 hours)", what were the samples in the experiment? Were the genes found that were known to be regulated by ABA or PIF?

The experiment that defined PIF-regulated genes in SD at the end of the night was performed by our laboratory (Martín et al., 2016), and the samples were 3-day old SD-grown seedlings, same conditions as the current work. This work suggested crosstalk of PIF and ABA pathways.

The experiment that defined ABA-responsive guard-cell specific genes by Wang et al. employed a novel protocol to isolate epidermal peels with guard cells as the only intact cell type. These epidermal peels were then used as the source of guard cell RNA for their microarray analysis comparing ABA-treated with solvent-treated, to identify ABA-responsive genes in GC. Next, they compared this gene set to ABA-responsive genes in whole leaves, to define a set of genes that are ABA-responsive and guard-cell specific (not detected in whole leaves) (906 genes, 515 induced and 384 repressed by ABA). The genes included genes known to be regulated by ABA, but the work did not include a comparison with PIF-regulated genes.

A clarification about the nature of the samples has been added to the text (lines 233 and 235-236).

15. "Interestingly, one of these genes (AT5G46240) encodes the KAT1 voltage-dependent

potassium channel predominantly expressed in GC (Fig. 4b)", AT5G46240 is KAT1? Pay attention to the accuracy of sentence description.

The sentence has been changed to "Interestingly, one of these genes (*AT5G46240*) is *KAT1*, encoding the voltage-dependent potassium channel *KAT1* predominantly expressed in GC (Fig. 4b)" (line 238-240).

16. *KAT1* is regulated by ABA and PIF, and whether qPCR or other assays have been conducted by yourselves. There is insufficient evidence that PIFs regulates *KAT1*.

Is *KAT1* expression unaffected by light in *pifq*?

We have conducted *KAT1* expression analyses shown in Fig. 4d, SI Fig. 7, and in new SI. Fig S6 (Col-0) and SI Fig. 12 (*KAT1OX*). We do not agree with the reviewer's view that there is insufficient evidence that PIFs regulate *KAT1*: Fig. 4d and SI Fig. 7 clearly show that (1) *KAT1* expression in Col-0 in SD is high in the dark at ZT0 and decrease upon exposure to light at ZT3, and therefore *KAT1* is a light-repressed gene; and (2) PIFs are necessary to promote *KAT1* expression in Col-0 in the dark at ZT0. These results agree with the decrease *KAT1* expression in Col-0 at ZT0 in LL, where PIFs do not accumulate, similar to *pifq*.

The results also show that the effect of light on *KAT1* expression seen in Col-0 is mediated by the PIFs, which accumulate in the dark (ZT0) but are degraded upon exposure to light resulting in a decrease in *KAT1* expression.

Do the results in Fig.4b and 4c come from the websites? Have they been repeatedly verified by yourselves?

Data shown in 4b and 4c come from the websites specified in the main text and legend of Fig. 4. We have not verified the experiment in 4b, but *KAT1* expression in guard cells and downregulation of *KAT1* expression after application of ABA are well reported in the literature (Leonhardt et al, Plant Cell 2004; Takahashi et al, Sci Signal 2013; Wang et al, BMC Genomics 2011). Regarding 4c, we have performed verification of *KAT1* expression in SD and LL in the time points most significant for stomata opening (ZT0 and ZT3), not only in Col-0 (as it is shown in 4c), but also in *pifq* and *aba2*. The results are shown in in Fig. 4d and SI Fig. 7.

In Fig.4d, why is there only a difference in 0 SD, and no significant difference between *aba2* and Col-0?

In agreement with Fig 4c, *KAT1* expression in Col-0 is high at the end of the night, drops after being exposed to light, and is maintained low in LL. Our results show that PIFs are necessary to induce the peak of *KAT1* expression at the end of the night, and therefore *KAT1* is a light-repressed gene. In contrast, ABA has been shown to regulate *KAT1* post-transcriptionally (Sutter et al., 2007), and our results do not support transcriptional regulation of *KAT1* by ABA levels at ZT0 and only marginally at ZT3 (Fig. 4d).

Would it be better to use *KAT1* protein level changes, or add ABA or light treatment?

We did not attempt to detect *KAT1* protein levels. *KAT1* protein has only been detected when overexpressed either transiently (Sutter et al., 2006 TPC; 2007 Curr Biol; Sieben et al., 2008, TPJ; Lebaudy et al., 2009 JBC;), or stably (Yenush laboratory, unpublished). Although it would be of great interest, detailed dynamics of the *KAT1* protein is beyond the scope of this work.

Regarding ABA changes, and as mentioned above, we are choosing to only assay the effect of the fluctuation in basal ABA levels in our experimental setup, instead of exogenously applied ABA. Regarding light treatment changes, we feel that one of the great advantages of our SD setup is the fluctuation in the light environment, covering dark and light and also time of day.

17. “These data suggest that endogenous basal ABA does not significantly repress PIF-mediated induction of *KAT1* expression at night, and might contribute to the repression of *KAT1* expression in SD upon exposure to light”, how to draw the conclusion?

The conclusion is based on the expression data shown in Fig 4d, where *KAT1* levels are not significantly affected in *aba2* mutant compared to Col-0 at ZT0, and are slightly more increased compared to Col-0 in ZT3. We believe these expression data using the ABA-deficient mutant *aba2* strongly support the conclusion mentioned by this reviewer.

18. “Taken together, we conclude that in SD PIFs are necessary to induce the expression of *KAT1*, a GC specific gene that encodes the inward-rectifying K⁺ channel driving the K⁺ uptake that leads to stomata opening upon activation by blue light”, would it be better to add red-light and blue-light treatment?

We only detected a small increase in stomata opening in response to red light (Fig. 3), which was similar in Col-0 and *pifq*. Therefore, we do not have any evidence of a PIF role for stomata opening under red light in the conditions used, and chose to leave the statement as it was.

19. CHIP-seq data showed binding of PIF1, PIF3 and PIF4 to the region, why chose PIF3 but not PIF1 or PIF4? Moreover, only a CHIP-qPCR experiment is not enough to convince that *KAT1* is directly downstream of PIF. For CHIP-qPCR, primer combinations should be better selected to identify binding sites.

We predicted that *KAT1*, with a clear CHIP-seq peak in PIF1 and PIF4, and also in PIF3 (albeit not significant given the background of the control experiment, shown in the light green track) was directly targeted by PIF3 as well. This was demonstrated by CHIP-qPCR, as shown in Fig.3b.

Our choice of primers spanned the 2 regions (P1 and P2) most likely bound by PIF3 in the promoter of *KAT1* based on (a) alignment with the CHIP-seq peak, and (b) presence of well-known putative PIF binding boxes (G-box and PBE, indicated in the gene diagram in Fig. 5a). In addition, a third primer pair was tested as control in the third exon of *KAT1* (P3).

Following also the suggestion by Reviewer 1, we are now showing specific binding of PIF3 to the P2 region by yeast one hybrid (SI Fig. 8). Compared to the plasmid expressing GAD alone, GAD-PIF3 was able to enhance binding to the P2 region in the *KAT1* promoter. Importantly, binding was lost in a mutated P2 region harboring a mutated version of the G-box motif, supporting direct binding to the *KAT1* promoter through the G-box motif in the P2 region. We could not detect specific binding of GAD-PIF3 to the P1 region, in accordance to our CHIP-qPCR data in Fig. 3b. We have updated the Methods section to include the yeast one-hybrid assay (lines 505-522). Results are shown in new Fig. S8 and discussed in the main text (lines 286-287).

20. Was the K⁺ changes measured in *kat1* and KAT1 OX? Are stomatal movements induced by K⁺ changes in guard cells? Just observing stomatal aperture is not enough to support your claim.

We believe that our conclusions regarding KAT1 are well supported by the genetic, phenotypic, molecular and expression data provided in the manuscript. Importantly, we have now added *pifqKAT1OX* lines that functionally complement the lack of stomatal opening phenotype of *pifq* (Fig. 6c, SI Fig. 10), providing conclusive support for the role of the PIF-KAT1 module in the regulation of stomata dynamics.

K⁺ changes in the *kat1* mutants were previously shown by Szyroki et al. (PNAS, 2001). Additionally, several assays have been done in heterologous systems expressing KAT1 (for example Clark MD et al., Nature 2020). These published results are in agreement with the well-established role of KAT1 as K⁺ channel. Triggering of stomatal movements by K⁺ changes in the guard cell is also well established in the literature (Lawson and Matthews, Annu Rev PS 2020). To our knowledge, K⁺ changes have not been measured KAT1-OX plants. Our results showing slightly more open stomata in KAT1-OX lines compared to Col-0 (Fig. 6b) might suggest that there is more K⁺ inflow leading to slightly more stomata opening.

21. Why discuss the similarity of PIF in regulating stomata movement and hypocotyl elongation? Whether it is more appropriate to discuss PIF and circadian rhythm?

Both responses (stomata movement and hypocotyl elongation) have in common that are maximal in short photoperiods (like SD) but greatly decreased in entrained seedlings (LL condition) (this work (Fig. 1d) for stomata, and Soy et al. 2014 for hypocotyl), indicating that they are not a direct output of the circadian clock. Indeed, it is the rhythmicity of PIF protein accumulation the underlying cause of rhythmic hypocotyl elongation in alternating day/night cycles, particularly those of SD (Soy et al., 2012 and PNAS 2016), and it is in this regard that we believe that it is very timely to draw comparison between stomata movements and hypocotyl elongation. In addition, because PRR clock proteins like TOC1 gate PIF activity to promote hypocotyl elongation (Soy et al, PNAS 2016; Zhu et al., Nat Comm 2016; Martín et al., Curr Biol 2018; Zhang et al., PNAS 2020), we include discussion on a possible role of the central circadian clock proteins PRRs in the regulation of stomata dynamics (lines 356-363).

22. “Interestingly, recent studies have reported a role for PIFs as negative regulators of stomata opening under red light. Although we did not detect this effect in our experiments in red light (Fig. 3)”, you did not detect this effect because of PIF knockout was used. PIF was degraded under red light, so there would be no difference between Col-0 and *pifq*, which does not mean that this phenomenon does not exist.

Because we did not detect a difference between Col-0 and *pifq* under red light, we could not suggest a role for PIFs under red light in our conditions. This is in contrast to what we observed under blue light, where the difference between Col-0 and *pifq* does support a role for PIFs in the blue light induction of stomata opening.

Possible explanations for the differences observed with other studies are provided in the Discussion (lines 367-374) as well as above in the response to Comment 8.

23. Why do you draw three models, including Fig. 7 and SI Fig. 5? Sum them up on a figure, if possible.

We feel that models in Fig. 7 exemplify the complexity of the guard cell with all the players involved in stomata pore opening (7a), whereas 7b focuses on the role of PIFs. We have, nevertheless, simplified model 7b following a comment by reviewer 1. Model in SI Fig. 11, on the other hand, capture the temporal dynamics across the 24h cycle, with the fluctuations in basal ABA and PIF levels. We believe the 3 models are complementary and will be useful to the readers, and have chosen to leave them as they were.

24. The overall association of ABA was not strong, and there was insufficient evidence for ABA involved in the regulation of this pathway.

We strongly disagree with this comment, given our genetic evidence provided with *aba2*, an ABA-deficient mutant, both at the phenotypic level and in the regulation of *KAT1* expression. Comments to the significance of our evidence for ABA involvement are provided in our response to Comments 9, 16, and 17.

25. At night, PIF accumulation promotes *KAT1* transcription, while endogenous ABA inhibits *KAT1* transcription. According to this statement, does *KAT1* channel increase every day. Whether there is another mechanism to achieve the dynamic balance of *KAT1*.

Because the abundance of PIF proteins is highly rhythmic during SD conditions, this can already provide a basis for the dynamic accumulation of *KAT1*. Superimposed to this regulation, fluctuations in endogenous ABA will determine the functionality of *KAT1* protein. Together, they gate *KAT1* activity to the beginning of each day. We captured this dynamic behavior in the model in SI Fig. 11.

Our results do not exclude other layers of regulation, which would require further analysis beyond the scope of this work.

26. How cotyledons were sampled to keep stomata constant, how many biological replicates were carried out. Check the full text, all experiments should be carried out in three or more biological replicates.

We apologize for not giving sufficient detail. As also explained in our response to reviewer 1, cotyledons were wet mounted in the dark with a dim green safelight. Then, images were taken with a Zeiss AxioPhot DP70 optical microscope using the same settings we used for the light samples. Illumination with the microscope light allowed observation and photography of the stomata in the mounted cotyledons. The imaging process was fast, which we deemed appropriate to keep the dark state of the stomata, as evidenced by the observed differences in stomata opening with the samples taken during the light time points.

In each experiment, 6 cotyledons of different seedlings were sampled. And for each cotyledon, multiple pictures were taken to obtained images of ~40-100 stomata. Three biological replicates were done with similar results, and one representative one is shown. For expression data, biological triplicates were assayed and graphs represent the average and variation of the mean.

We have revised the Methods and Figure Legends accordingly to add this additional information.

Reviewer #4 (Remarks to the Author):

In the present manuscript, Rovira and colleagues provide an interesting new look at possible mechanisms that regulate rhythmic stomatal movements during the night-to-day transition. The authors provide evidence that PIF transcription factors are required for stomatal opening in the morning as well as blue light-induced stomatal opening. They also show that PIFs are required for accumulation of guard cell-specific K⁺ channel KAT1 at night. Furthermore, the authors provide evidence that PIF3 binds to the KAT1 promoter by ChiP-qPCR analysis. From these results, the authors conclude that PIF accumulation at night promotes KAT1 expression, which ensures stomatal opening in the morning.

Major concerns

1) Genetic interaction of PIF and KAT1 expression. The authors provided biochemical evidence that PIF3 directly binds to the KAT1 promoter by ChiP-qPCR (Fig. 5b).

Thank you for your positive comments.

If the inhibition of stomatal opening seen in the *pifq* mutant is due to decreased expression of KAT1, then *pifq* should exhibit stomatal opening by introduction of KAT1OX. Otherwise, the entire argument for PIF-induced KAT1 expression and stomatal opening are flawed. These data would strengthen their conclusion.

Thank you for this important suggestion. We have generated the suggested *pifqKAT1OX* line. As explained above in our response to Reviewer 2, we have generated *pifqKAT1OX* mutants to test their genetic interaction. As shown in new Fig.6c and new SI Fig. 10, KAT1 overexpression was able to completely restore the stomatal opening phenotype in *pifq*, both under SD conditions (new Fig. 6c) and in response to blue light (new SI Fig. 10). These results support the essential function of KAT1 downstream of PIFs in the regulation of stomata opening, and strengthen the conclusion that inhibition of stomatal opening in *pifq* is due to decreased *KAT1* expression.

These new important data are discussed in the text in lines 308-313.

2) The authors showed that KAT1OX in Col-0 exhibited larger stomatal opening at ZT0 and ZT3 and slower stomatal closure at ZT6 compared to Col-0 (Fig. 6b). How do they explain stomatal closure in KAT1OX at ZT6? Given that KAT1 is constitutively expressed in KAT1OX, their stomata should open during light irradiation.

Our finding that the ABA-deficient mutant *aba2* does not close stomata between ZT3-ZT9 as sharply as Col-0 (Fig. 1a) suggests that ABA (together with other unknown factor(s)) might be important to trigger stomata closing during this time, as indicated in the text (lines 179-180). Because ABA has been shown to prevent KAT1 accumulation through post-transcriptional mechanisms such as KAT1 endocytosis and sequestration (Sutter et al., Curr Biol 2007) (line 337), this could be a mechanism by which stomata could close at dusk even if KAT1 is overexpressed. Interestingly, in LL conditions, our finding that *aba2* stomata remain open at dusk (SI Fig. 5), suggests that the unknown factor(s) acting together with ABA might not accumulate or be as active in LL. We therefore anticipated that in these LL conditions, KAT1OX would display a more continuous open stomata phenotype. Indeed, this is what we found, which is now included as new SI Fig. S9.

3) Furthermore, if the lack of stomatal opening in Col-0 in LL conditions is attributed to decreased expression of *KAT1*, does *KAT1OX* show stomatal opening in LL?

As mentioned above, stomata opening in *KAT1OX* in LL is significantly increased compared to Col-0 LL. This is now included as new SI Fig. 9 and discussed in lines 302-308.

4) Previous studies have shown that phytochromes positively regulate light-induced stomatal opening (Wang et al. 2010, Mol Plant, etc). On the other hand, according to the model proposed by the authors (Fig. 7b), phytochromes negatively regulate *KAT1* expression and stomatal opening via PIF degradation. The authors need to assay the expression of *KAT1* (e.g. Fig. 4d) and stomatal opening (e.g. Fig. 1d) in phytochrome mutants. These data would expand our understanding of phytochrome-mediated control of stomatal movements.

We have conducted the suggested experiment. As previously reported, stomata opening in *phyAB* mutants was reduced. Interestingly, we did not find *KAT1* expression to be significantly affected in *phyAB* mutants (see Figure below). Because PIF abundance is increased in the absence of *phyAB* (we have shown this increase under the same SD conditions for PIF3 in Soy et al., 2012), we conducted an experiment with PIF-OX mutants and found that increased PIF abundance does not result in higher *KAT1* expression (new SI Fig. 4), in accordance to the *KAT1* levels obtained in *phyAB*.

Together, these results suggest that phytochromes can affect stomata dynamics through the regulation of factors or pathways other than PIFs. One possibility would be for example through the impact on photosynthesis (Yang et al., PNAS 2016). We have adjusted the model in 7b and the legend to reflect other pathways through which phytochromes can affect stomata opening.

Minor concerns

1) In association with Major concern 2, I recommend the authors to provide *KAT1* expression in *KAT1OX* under stomatal experimental conditions.

KAT1 expression in the *KAT1OX* line in our experimental condition is now included as SI Fig. 12.

2) Fig. 1a, 1b, 1d, 4, 6a, and 6b – please include light/dark bar.

We appreciate this suggestion. Although this was our original idea, given that some graphs include more than 1 light regime (1d, 4c, 4d) we instead chose to include a figure with the description of the different light regimes used (SI Fig. 1) to prevent confusion. We have, in

any case, indicated in each panel the light regime used: SD, LL, and also DD in Supplemental Figures.

3) No information is provided about any differences in stomatal size among Col-0, aba2, pifq, kat1, and KAT1OX. Since the authors reported stomatal pore area and size, this information is needed to validate the data.

We are now providing measurements of pore length for Col-0, aba2 and pifq (new SI Fig 2). Results show that opening of pore area is due mostly to variation in width, rather than length.

4) Page 4, GHR1 has been reported to be kinase-dead (Sierla et al. 2018). It is not appropriate to quote in this context.

Thank you for spotting this inaccuracy. The quote has been now removed.

5) Page 4 - OST2 - AHA1/OST2.

Corrected (line 77).

6) Page 4 -Na⁺/H⁺ - superscript.

Corrected (line 79).

Reviewer #1 (Remarks to the Author):

I believe that the authors have reasonably and adequately addressed previous concerns/comments. I have no further questions.

Reviewer #2 (Remarks to the Author):

Comparing stomatal opening in *pifQ* versus *pifQ KAT1-OX* lines, the authors have shown conclusively that *KAT1* acts downstream of PIFs and that *KAT1* indeed appears to be the major component mediating PIF-induced stomatal opening in the early morning. They have thereby addressed a major concern raised by myself and other reviewers.

The observations that neither PIF overexpression nor phytochromes (which are known to trigger PIF degradation in the light) affect *KAT1* expression are slightly puzzling and suggest other regulators are likely to contribute to *KAT1* upregulation. As the authors explain, post-translational regulation of *KAT1* may add another layer in controlling *KAT1*-dependent stomatal opening, but this was not addressed experimentally in this study.

Overall, the conclusions drawn are valid, although I maintain that they are of interest mainly to a specialised audience.

Reviewer #3 (Remarks to the Author):

The article found that PIF affects rhythmic stomata movements in daily dark/light cycles by regulating K⁺ channel gene *KAT1*, providing novel insights into the regulation of stomatal opening by blue light and PIF. While this represents a compelling discovery, we feel that there are still some problems that are not enough to publish in NC. One notable concern is the reliance on indirect evidence for many experimental conclusions, exemplified by experiments involving ABA (abscisic acid) content, and direct evidence may be more revealing. And, the direct regulation of *KAT1* transcription by PIFs only verified *PIF3*, which was insufficient to explain the regulatory function of PIFs on *KAT1*, and so on.

Reviewer #4 (Remarks to the Author):

I have previously reviewed the manuscript as Reviewer 4, and I am pleased to note that the authors have addressed the concerns raised in the initial review convincingly. Additionally, I have some further comments on the manuscript.

1. The authors argued that phytochrome-mediated PIF degradation represses *KAT1* induction during the day. However, data in the response letter submitted by the authors show a decrease in the expression levels of *KAT1* at ZT3 in *phyAB*. This additional data suggests that the repression of *KAT1* induced by light is independent of phytochromes, which contradicts the model presented in Fig. 7. How do the authors explain this contradiction?

Did the authors confirm that PIF does not degrade in the guard cells of the *phyAB* mutant even when exposed to light? Even though PIF degradation is lacking, is there a decrease in *KAT1* expression at ZT3 in the *phyAB* mutant? If so, the stomatal closure during the day cannot be explained by the downregulation of *KAT1* through phytochrome-mediated PIF degradation.

2. In the additional data presented by the authors, in the *phyAB* mutant, despite *KAT1* expression being comparable to the wild type, stomata remain closed. Considering that phytochromes control the expression of thousands of genes, I understand the potential for other regulatory mechanisms beyond PIF and *KAT1* in the control of stomatal movements by phytochromes, as the authors suggest.

However, I believe it would be fair for the authors to present the results of stomatal movements in the *phyAB* mutant, as demonstrated in the additional data, and discuss the potential factors contributing to the contradiction between the expression of *KAT1* and the stomatal movements in

the main text. Otherwise, it may lead to confusion among readers regarding the control of stomatal movements by phytochromes.

3. The authors suggest post-transcriptional regulation of KAT1 abundance contributes to stomatal closure during the day in *KAT1OX SD* and *pifqKAT1OX SD*. While the authors discuss the potential role of ABA in regulating the KAT1 accumulation in lines 336-337, I recommend that the authors include information about the observed daytime stomatal closure phenotypes in *KAT1OX SD* and *pifqKAT1OX SD* in this passage to improve reader comprehension.

We thank again the reviewers for their comments. We have now addressed the last concerns, and the newly revised manuscript has been adjusted accordingly. We hope you will agree that the manuscript is now ready for publication.

Below are the last comments received followed by our answers:

REVIEWER COMMENTS

Reviewer #1 (Remarks to the Author):

I believe that the authors have reasonably and adequately addressed previous concerns/comments. I have no further questions.

Reviewer #2 (Remarks to the Author):

Comparing stomatal opening in *pifQ* versus *pifQ KAT1-OX* lines, the authors have shown conclusively that *KAT1* acts downstream of PIFs and that *KAT1* indeed appears to be the major component mediating PIF-induced stomatal opening in the early morning. They have thereby addressed a major concern raised by myself and other reviewers.

The observations that neither PIF overexpression nor phytochromes (which are known to trigger PIF degradation in the light) affect *KAT1* expression are slightly puzzling and suggest other regulators are likely to contribute to *KAT1* upregulation. As the authors explain, post-translational regulation of *KAT1* may add another layer in controlling *KAT1*-dependent stomatal opening, but this was not addressed experimentally in this study.

We agreed with this Reviewer and Reviewer #4 that our finding that increased PIF levels (either by PIF overexpression or in a phytochrome mutant lacking *phyA* and *phyB*) did not have an effect on *KAT1* expression was slightly puzzling and important to address in the final revision of our work.

We hypothesized that transcriptional repression of PIF activity could play a role in the homeostasis of PIF/*KAT1*, preventing increased *KAT1* expression. In fact, we previously showed a role for the clock components PSEUDO-RESPONSE REGULATORS (PRRs) as negative regulators of PIF activity under day-night cycles in the regulation of hypocotyl elongation (*Circadian Waves of Transcriptional Repression Shape PIF-Regulated Photoperiod-Responsive Growth in Arabidopsis*. Martín G et al., *Curr Biol*. 2018). Analyses of PRR ChIP-seq experiments performed by Eva Farré and colleagues (*Liu et al.*, 2016) indeed revealed that PRR5 directly targets *KAT1*. We have now tested *KAT1* expression in a *prp5*-mutant under our diurnal conditions and have found elevated expression of *KAT1* (see new SI Fig. 14). Although further experiments beyond the scope of this work are necessary to fully understand the interplay of the clock protein and transcriptional repressor PRR5 in the regulation of *KAT1* expression and stomata dynamics, this new result together with previous published data strongly suggest that PIF activity in the regulation of *KAT1* expression is under regulation of PRR5 to prevent over-induction of *KAT1*. We think that this multi-layer

regulation reflects the importance of tight regulation of *KAT1* expression and stomata opening to optimize seedling growth.

We thank this reviewer for the suggestion to include additional experimental evidence, which we believe has helped us improve our manuscript.

The *phyAB* data is now included in the manuscript (New SI Fig 5), and the new *prp5* data has been added as SI Fig 14. The text has been updated accordingly.

Overall, the conclusions drawn are valid, although I maintain that they are of interest mainly to a specialised audience.

We cannot agree with this statement. Our findings describe a new mechanism by which plants exquisitely adjust stomata dynamics to allow gas exchange regulation in accordance to time of day and environmental conditions. We believe the novelty and impact of this work are of interest to a broad audience.

Reviewer #3 (Remarks to the Author):

The article found that PIF affects rhythmic stomata movements in daily dark/light cycles by regulating K⁺ channel gene *KAT1*, providing novel insights into the regulation of stomatal opening by blue light and PIF. While this represents a compelling discovery, we feel that there are still some problems that are not enough to publish in NC. One notable concern is the reliance on indirect evidence for many experimental conclusions, exemplified by experiments involving ABA (abscisic acid) content, and direct evidence may be more revealing. And, the direct regulation of *KAT1* transcription by PIFs only verified PIF3, which was insufficient to explain the regulatory function of PIFs on *KAT1*, and so on.

To address the remaining concerns, we have now included evidence that PIF4 directly binds to the *KAT1* promoter, similar to PIF3 (Fig 5c), as predicted by the ChIP-seq data shown in Fig 5a. We have updated the text accordingly.

We are now also including an experiment done with exogenously applied ABA under our diurnal conditions (New SI Fig. 9). Exogenously applied ABA, as expected from previous reports in other growth systems, (a) significantly repressed *KAT1* expression, and (b) prevented morning stomata opening.

These results further support the significance of our findings regarding the delicate regulation of stomata dynamics by the balance of PIF and endogenous ABA levels to precisely time *KAT1* expression and accumulation at dawn, necessary for morning stomata opening under well-watered conditions. We have updated the text accordingly.

We thank the reviewer for helping us improve our manuscript.

Reviewer #4 (Remarks to the Author):

I have previously reviewed the manuscript as Reviewer 4, and I am pleased to note that the authors have addressed the concerns raised in the initial review convincingly. Additionally, I have some further comments on the manuscript.

1. The authors argued that phytochrome-mediated PIF degradation represses KAT1 induction during the day. However, data in the response letter submitted by the authors show a decrease in the expression levels of KAT1 at ZT3 in phyAB. This additional data suggests that the repression of KAT1 induced by light is independent of phytochromes, which contradicts the model presented in Fig. 7. How do the authors explain this contradiction?

Did the authors confirm that PIF does not degrade in the guard cells of the phyAB mutant even when exposed to light? Even though PIF degradation is lacking, is there a decrease in KAT1 expression at ZT3 in the phyAB mutant? If so, the stomatal closure during the day cannot be explained by the downregulation of KAT1 through phytochrome-mediated PIF degradation.

Thank you for this comment. *KAT1* expression in phyAB is indeed decreased at ZT3 similarly to WT (data are now included as New SI Fig. 5).

As explained in our response to Reviewer #2, we agreed with this Reviewer and Reviewer #2 that our finding that increased PIF levels (either by PIF overexpression or in a phytochrome mutant lacking phyA and phyB) did not have an effect on *KAT1* expression was slightly puzzling and important to address in the final revision of our work.

Based on our previous findings showing that the clock components PSEUDO-RESPONSE REGULATORS (PRRs) act as negative regulators of PIF activity under day-night cycles in the regulation of hypocotyl elongation (*Circadian Waves of Transcriptional Repression Shape PIF-Regulated Photoperiod-Responsive Growth in Arabidopsis*. Martín G et al., *Curr Biol*. 2018, we hypothesized that transcriptional repression of PIF activity could play a role in the homeostasis of PIF/KAT1, preventing increased *KAT1* expression. Analyses of PRR ChIP-seq experiments performed by Eva Farré and colleagues (*Liu et al., 2016*) indeed revealed that PRR5 directly targets *KAT1*. We have now tested *KAT1* expression in a *prp5*- mutant under our diurnal conditions and have found elevated expression of *KAT1* (see new SI Fig. 14). Although further experiments beyond the scope of this work are necessary to fully understand the interplay of the clock protein and transcriptional repressor PRR5 in the regulation of *KAT1* expression and stomata dynamics, this new result together with previous published data strongly suggest that PIF activity in the regulation of *KAT1* expression is under regulation of PRR5 to prevent over-induction of *KAT1*. We think that this multi-layer regulation reflects the importance of tight regulation of *KAT1* expression and stomata opening to optimize seedling growth.

The new *prp5* data has been added as SI Fig 14 and the text has been updated accordingly.

2. In the additional data presented by the authors, in the phyAB mutant, despite KAT1 expression being comparable to the wild type, stomata remain closed. Considering that phytochromes control the expression of thousands of genes, I understand the potential for other regulatory mechanisms beyond PIF and KAT1 in the control of stomatal movements by phytochromes, as the authors suggest.

However, I believe it would be fair for the authors to present the results of stomatal movements in the phyAB mutant, as demonstrated in the additional data, and discuss the

potential factors contributing to the contradiction between the expression of KAT1 and the stomatal movements in the main text. Otherwise, it may lead to confusion among readers regarding the control of stomatal movements by phytochromes.

We thank the reviewer for this suggestion. We are now including the phyAB data in the manuscript as New SI Fig. 5. We have now adjusted the text accordingly to describe and discuss these findings, together with the new prr5 results mentioned in the previous point.

3. The authors suggest post-transcriptional regulation of KAT1 abundance contributes to stomatal closure during the day in KAT1OX SD and pifqKAT1OX SD. While the authors discuss the potential role of ABA in regulating the KAT1 accumulation in lines 336-337, I recommend that the authors include information about the observed daytime stomatal closure phenotypes in KAT1OX SD and pifqKAT1OX SD in this passage to improve reader comprehension.

Following the recommendation, we are now including the suggested information in the discussion passage. We agree that it improves reader comprehension.

Reviewer #4 (Remarks to the Author):

I am pleased to see that the authors have responded convincingly to the concerns raised in the second review. I have no additional comments on the manuscript.